# Effect of tip spacing, thrust coefficient and turbine spacing in multi-rotor wind turbines and farms

Niranjan S. Ghaisas[1,4], Aditya S. Ghate[2], and Sanjiva K. Lele[1,2,3]

[1]Center for Turbulence Research, Stanford University, Stanford, CA 94305, USA.
[2]Department of Aeronautics and Astronautics, Stanford University, Stanford, CA 94305, USA.
[3]Department of Mechanical Engineering, Stanford University, Stanford, CA 94305, USA.
[4]Department of Mechanical and Aerospace Engineering, Indian Institute of Technology Hyderabad, Kandi, Telangana 502285, India.

**Correspondence:** Niranjan S. Ghaisas (nghaisas@iith.ac.in)

**Abstract.** Large eddy simulations (LES) are performed to study the wakes of a multi-rotor wind turbine configuration comprising of four identical rotors mounted on a single tower. The multi-rotor turbine wakes are compared to the wake of a conventional turbine comprising of a single rotor per tower with the same frontal area, hub height and thrust coefficient. The multi-rotor turbine wakes are found to recover faster, while the turbulence intensity in the wake is smaller, compared to the wake of the conventional turbine. The differences with the wake of a conventional turbine increase as the spacing between the tips of the rotors in the multi-rotor configuration increases. The differences are also sensitive to the thrust coefficients used for all rotors, with more pronounced differences for larger thrust coefficients. The interaction between multiple multi-rotor turbines is contrasted with that between multiple single-rotor turbines by considering wind farms with five turbine units aligned perfectly with each other and with the wind direction. Similar to the isolated turbine results, multi-rotor wind farms show smaller wake losses and smaller turbulence intensity compared to wind farms comprised of conventional single-rotor turbines. The benefits of multi-rotor wind farms over single-rotor wind farms increase with increasing tip spacing, irrespective of the axial spacing and thrust coefficient. The mean velocity profiles and relative powers of turbines obtained from the LES results are predicted reasonably accurately by an analytical model assuming Gaussian radial profiles of the velocity deficits and a hybrid linear-quadratic model for merging of wakes. These results show that a larger power density can be achieved without significantly increased fatigue loads by using multi-rotor turbines instead of conventional, single-rotor turbines.

*Copyright statement.* TEXT

## 1 Introduction

Wind energy is among the fastest growing sources of renewable energy worldwide. Understanding and mitigating the deleterious effects of interactions between wakes of multiple turbines is critical for efficient utilization of the wind resource. In large wind farms, the wake interactions can limit the power density, or the power extracted per unit land area. The turbulent

wake interactions also determine fatigue loads on downstream turbines, which has a direct bearing on the levelized cost of energy. Previous work has shown that wake losses are closely tied to wind farm layout parameters such as inter-turbine spacing (Meyers and Meneveau, 2012; Yang et al., 2012), alignment between columns and the wind direction (Stevens et al., 2014a; Ghaisas and Archer, 2016), horizontal staggering between adjacent rows (Archer et al., 2013) and vertical staggering of similar

or dissimilar turbines (Vasel-Be-Hagh and Archer, 2017; Xie et al., 2017; Zhang et al., 2019).

The idea of mounting multiple rotors per tower has been explored in recent years (Jamieson and Branney, 2012, 2014; Chasapogiannis et al., 2014; Ghaisas et al., 2018; van der Laan et al., 2019; Bastankhah and Abkar, 2019). For example, Jamieson and Branney (2012) pointed out that the scaling laws for power and weight with the diameter of a turbine (the 'square-cube law') pose a challenge to upscaling the design of current single-rotor turbines to very large systems, but make multi-rotor

turbines an attractive alternative. Structural considerations with designing a 20 MW multi-rotor system were investigated in Jamieson and Branney (2014). Their results suggested that for a 45-rotor 20 MW system, the benefits due to reduced rotor and drive train costs would outweigh potential challenges associated with a more complicated tower structure. Chasapogiannis et al. (2014) studied the aerodynamics of a 7-rotor system, with the tips of the blades of adjacent rotors spaced 0.05 diameters apart. Interference due to adjacent rotors was found to lead to approximately 3 % increase in power, while about 2 % increase

in the blade loading amplitude was observed.

Analysis of the wake of a 4-rotor turbine was carried out in our previous work (Ghaisas et al., 2018) using large eddy simulation (LES). It was shown that the multi-rotor turbine wakes recover faster compared to wakes of an equivalent single-rotor turbine. The turbulent kinetic energy added due to multi-rotor turbines was also lesser than that due to an equivalent single-rotor turbine. Wind farms comprising of five aligned turbines spaced four diameters apart were also considered in this

study. The potential for reduced wake losses as well as reduced fatigue loads was clearly pointed out.

The results for the wake of an isolated turbine were confirmed recently in van der Laan et al. (2019) using a combination of field observations and numerical simulations. van der Laan et al. (2019) also studied the aerodynamics of individual and combined rotors. It was found that rotor interaction can lead to an increase of up to 2 % in the power generation, similar to that reported in Chasapogiannis et al. (2014). Isolated multi-rotor turbines were studied in detail in van der Laan et al. (2019),

and potential benefits in multi-rotor wind farms were discussed. Bastankhah and Abkar (2019) also studied isolated multi-rotor wind turbine wakes and found similar wake recovery characteristics. Multi-rotor configurations other than the 4-rotor configuration studied in the present paper and elsewhere were considered. The effect of number and direction of rotation of the individual rotors on the rate of wake recovery was also studied, and was found to be negligible by Bastankhah and Abkar (2019).

Interactions between several multi-rotor wind turbines arranged in a $4 \times 4$ grid were studied using several Reynolds-Averaged Navier Stokes (RANS) simulations and one large eddy simulation (LES) in van der Laan and Abkar (2019). The annual energy production of multi-rotor wind farms was found to be $0.3 - 1.7\%$ larger compared to that of equivalent single-rotor wind farms. The benefit was confined to the first downstream turbine row and for cases where the wind direction was fully aligned with the turbine columns. This discrepancy with the results of Ghaisas et al. (2018) can be attributed to the large tip spacings

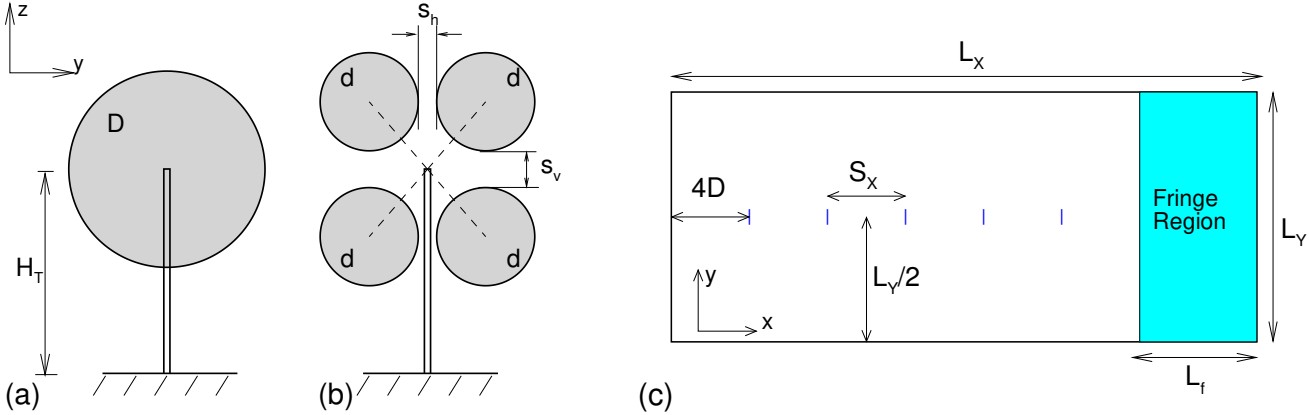

**Figure 1.** Schematic of (a) conventional 1-rotor turbine and (b) 4-rotor turbines. Tower height $H_T$ is identical for both turbines. Diameters are related by $D = 2d$. Spacing between tips is $s_h$ in horizontal and $s_v$ in vertical. (c) Schematic of the computational domain in plan view, not to scale. Blue lines denote turbine locations.

considered in Ghaisas et al. (2018). In the present work, we study more realistic tip spacings, and observe consistent qualitative and quantitative trends with the results of van der Laan and Abkar (2019).

In this paper, we extend our previous work (Ghaisas et al., 2018) by considering a larger number and range of multi-rotor wind turbine and farm design parameters. A schematic of the multi-rotor turbine considered here is shown in Fig. 1(b). Four rotors with identical diameters, $d$, are mounted on a tower with height $H_T$ (Fig. 1b). The tips of the rotors are separated by $s_h$ and $s_v$ in the horizontal and vertical, respectively. As a result, the rotors are centered at $H_T \pm (s_v + d)/2$, and the mean hub-height is $H_T$. The multi-rotor configuration (henceforth referred to as 4-rotor turbine) is compared to a conventional turbine with a single rotor (referred to as 1-rotor turbine) with diameter $D = 2d$ per tower with height $H_T$ (Fig. 1a). The total frontal rotor area is $\pi D^2/4$ in each case.

The primary aim of this paper is to quantify the benefits associated with the wakes of multi-rotor turbines for a wide range of tip spacings, thrust coefficients and inter-turbine spacings using LES. A second aim is to develop an analytical modeling framework, combining elements from previously published studies, and to evaluate its ability to predict the mean velocity profiles in the wakes of multi-rotor wind farms. This study differs from that of van der Laan et al. (2019) mainly in the manner in which the undisturbed inflow profiles are imposed. The inflow in van der Laan et al. (2019) is a logarithmic profile corresponding to the neutrally stratified atmospheric surface layer, with an effectively infinite boundary layer height, while an ABL with a finite height is used as the inflow in the present study. Three levels of turbulence intensity at the hub-height were considered in van der Laan et al. (2019), while all cases in the present study have a fixed turbulence intensity. Pitch and torque controllers were adopted in the simulations of van der Laan et al. (2019), which produced realistic power curves over the entire region of operation of the single-rotor and multi-rotor turbines. In the present study, a constant thrust coefficient is imposed, which is a reasonably accurate representation of a turbine operating in 'Region II' of the power curve (Stevens et al., 2014a).

This paper is organized as follows. The LES methodology, details of the simulations and the analytical framework are described in Sect. 2. Results of isolated 4-rotor turbines are described in Sect. 3, while results of wind farms comprised of 4-rotor turbines are described in Sect. 4. In each case, LES results are presented followed by predictions of the analytical modeling framework. Sect. 5 presents a brief summary and the conclusions.

## 2 Numerical Methodology

### 2.1 Simulation Framework

The LES-filtered incompressible Navier-Stokes equations are solved on a structured uniform Cartesian mesh using Fourier-collocation in $x$ and $y$ directions, sixth-order staggered compact finite-differences in the $z$ direction and a total variation diminishing (TVD) fourth-order Runge-Kutta time-stepping scheme. Non-periodicity is imposed in the $x$ direction using a fringe region technique (Nordström et al., 1999). Partial dealiasing is achieved by applying the 2/3 rule in $x, y$ and the use of skew-symmetric form for the convective terms in the $z$ direction. The governing equations and numerical discretization details may be found in Ghate and Lele (2017) (Appendix A). The effect of sub-filter scales is modeled using the Anisotropic Minimum Dissipation (AMD) model (Rozema et al., 2015). Wind turbine forces are modeled as momentum sinks using the actuator drag-disk model (Calaf et al., 2010). The turbine forces in the LES are defined in terms of the disk-averaged velocity and a 'local thrust coefficient', $C_T'$. The local thrust coefficient (assuming validity of the inviscid actuator-disk theory) is related to the nominal thrust coefficient, $C_T$, through the relation $C_T = 16 C_T' / \left(C_T' + 4\right)^2$, or equivalently, through the relations $C_T' = C_T/(1-a)^2$ and $a = \left(1 - \sqrt{1-C_T}\right)/2$, where $a$ is the axial induction factor. Algebraic wall models based on the Monin-Obukhov similarity theory are used to specify the shear stresses at the bottom wall. Viscous stresses in the rest of the domain are smaller than the sub-filter scale stresses by around 8-10 orders of magnitude and, hence, are neglected in these simulations. The code has been validated over several previously published studies (Ghate and Lele, 2017; Ghaisas et al., 2017; Ghate et al., 2018).

### 2.2 Cases Simulated

Half-channel (HC) simulations are carried out using the concurrent precursor-simulation methodology (Stevens et al., 2014b) on domains of length $L_x, L_y, L_z$ in the three coordinate directions. A schematic of the simulation domain is shown in Fig. 1(c). All simulations use $(L_y, L_z) = (\pi/2, 1) H$, while $L_x = \pi H$ or $1.25\pi H$, depending on the case. Here $H$ is the height of the half-channel. The flow in the 'precursor' simulation is driven by a constant imposed pressure gradient, $-u_*^2/H$, where $u_*$ is the friction velocity at the bottom wall. The HC configuration is used as a model for the neutrally-stratified atmospheric boundary layer (ABL) with the Coriolis forces neglected (Stevens et al., 2014a; Calaf et al., 2010), and we use the terms HC and ABL interchangeably. The surface roughness height at the bottom wall is $z_0 = 10^{-4}H$. This corresponds to rough land, and has been used in previous wind turbine studies (Calaf et al., 2010). The turbulence intensity at a typical hub height of $0.1H$ is approximately $8\%$. All results are normalized using scales $H$ and $u_*$, with typical values $H = 1000$ m, $u_* = 0.45$ m/s.

**Table 1.** Suite of isolated turbine (sets IT*) and wind farm (sets WF*) simulations. Domain lengths are non-dimensionalized by height $H$, with label D1 denoting $(\pi \times \pi/2 \times 1)$ and DA denoting $(1.25\pi \times \pi/2 \times 1)$. Grid sizes shown are for 'main' domain. Equal number of grid points are additionally required for the 'precursor' domain in each case. Labels G1, G2, G3, G4 and G5 denote grids of sizes $192 \times 96 \times 128$; $256 \times 128 \times 160$; $320 \times 160 \times 200$; $384 \times 192 \times 256$ and $512 \times 256 \times 320$; respectively. G2A denotes a grid with $320 \times 128 \times 160$ points. Axial spacing is undefined for isolated turbine simulations. Local thrust coefficients are $C_T' = 1, 4/3, 2$, corresponding to nominal $C_T = 0.64, 0.75, 8/9$, respectively.

| Set | Domain | Grid | Tip Spacing, $s_h/d = s_v/d = s/d$ | Thrust Coefficient, $C_T'$ | Axial Spacing, $S_X$ |
|---|---|---|---|---|---|
| IT1-s | D1 | G1 | 1-Rot, 0.05 | 4/3 | - |
| IT2-s | D1 | G2 | 1-Rot, 0.0, 0.05, 0.1, 0.2, 0.25, 0.5, 1.0 | 4/3 | - |
| IT3-s | D1 | G3 | 1-Rot, 0.05 | 4/3 | - |
| IT4-s | D1 | G4 | 1-Rot, 0.05 | 4/3 | - |
| IT5-s | D1 | G5 | 1-Rot, 0.05 | 4/3 | - |
| IT2-$C_T'$ | D1 | G2 | 1-Rot, 0.1 | 1.0, 2.0 | - |
| WF2-$C_T'$ | D1 | G2 | 1-Rot, 0.1, 0.25, 0.5 | 1.0, 4/3, 2.0 | 4D |
| WF2-SX | DA | G2A | 1-Rot, 0.1, 0.25, 0.5 | 4/3 | 5D, 6D |

Precursor simulations (without turbines and with streamwise periodicity) are carried out first for 50 time units (1 time unit = $H/u_*$), so as to achieve a fully-developed statistically stationary state. These velocity fields are then used to initialize the 'precursor' and 'main' simulation domains. Turbines are introduced in the 'main' domain, and a portion of this domain, of length $L_f = 0.15L_x$, is forced with the velocity field from the 'precursor' domain at each time step. Simulations in this concurrent precursor-simulation mode are carried out for a further 20 time units, with time-averaging performed using samples stored every 10 time steps over the last 12 time units. For the typical values of $H$ and $u_*$ mentioned above, this corresponds to approximately 12.3 hours of simulations with turbines, out of which statistics are collected over approximately 7.4 hours.

The suite of simulations carried out is listed in Table 1. In the first set of simulations (IT1, IT2, IT3), isolated turbines are simulated with a baseline 1-rotor configuration with $D = 0.1H$, and a baseline 4-rotor configuration with $d = 0.05H$ and $s_h = s_v = s = 0.05d$. Six additional (set IT2) isolated 4-rotor turbine simulations are carried out with varying $s$ to study the effect of tip spacing in the 4-rotor configuration. The thrust coefficient is fixed for this first set of simulations. In the second set (IT2-$C_T'$), four isolated turbine simulations are carried out to study the effect of varying thrust coefficient. In the third set of simulations (sets WF*), a line of five 1-rotor turbines separated by a distance $S_X$ in the streamwise direction is compared to a similar configuration with a line of five 4-rotor turbines separated by $S_X$ in the streamwise direction. A total of 20 wind farms are simulated, considering different combinations of $S_X$, $C_T'$ and $s$. The same thrust coefficient is used for all rotors

in one simulation. All isolated turbines, and the most upstream turbine in the five-turbine cases, are located at $x = 0$, where the domain inlet is at $x = -4D$. The turbine towers are located at $y = Ly/2$ in the spanwise direction and the tower height is $H_T = 0.1H$ for all turbines. The domain size in the x-direction is increased to $1.25\pi$ to accommodate larger axial spacings for the cases with $S_X = 5D$ or $6D$.

Field measurements and simulations reported in van der Laan et al. (2019) show that the bottom pair of rotors has a slightly larger thrust coefficient than the top pair of rotors. However, for simplicity, the same thrust coefficient is used for all rotors in one simulation. The methodology of keeping thrust forces identical across all rotors of the multi-rotor turbine was adopted by van der Laan et al. (2019) as well in the part of their study that focused on comparing wakes of multi-rotor and single-rotor turbines. The effect of variable operating conditions for the top and bottom pairs of rotors can be studied systematically in the

future. Finally, the appropriateness of considering a single-rotor turbine with the same total frontal area, thrust coefficient and mean hub height as that of the multi-rotor turbine is evaluated in Appendix B.

## 2.3   Analytical Model

An analytical modeling framework based on the model by Bastankhah and Porté-Agel (2014) is evaluated for the multi-rotor configuration in this paper. The model assumes that the velocity deficit in the wake decays in the streamwise ($x$) direction, and

follows a Gaussian profile in the radial directions. The deficit due to turbine rotor $i$ located at $(x_i, y_i, z_i)$ at a downstream point $(x, y, z)$ is given as

$$\frac{\Delta \bar{u}_i(x, y, z)}{\bar{u}_{up}(z)} = C(x) \times \exp\left(-\frac{(y - y_i)^2 + (z - z_i)^2}{2\left(k^*(x - x_i) + \sigma_0\right)^2}\right), \tag{1}$$

$$C(x) = \left(1 - \sqrt{1 - \frac{C_T}{8\left(k^*(x - x_i)/d_0 + \sigma_0/d_0\right)^2}}\right), \tag{2}$$

for $x > x_i$. The length scale $d_0$ equals $D$ for 1-rotor and $d$ for 4-rotor cases. The argument of the square-root in eq. (2) is set to

zero whenever it is less than zero, which happens very close to the turbines.

    The combined effect of multiple turbine rotors has been modeled in the past using several empirical techniques. Primary among these are addition of velocity deficits (implying linear addition of momentum deficit), square-root of sum of squares (implying addition of kinetic energy deficit; also termed as quadratic merging), and considering the largest deficit to be dominant. In this study, a hybrid between the first two approaches is found to give best results. Appendix A presents brief comments

justifying the hybrid approach. The hybrid approach involves linear merging of wakes originating at the same $x$ location, with quadratic merging of wakes originating at different $x$ locations. This can be written as

$$\Delta \bar{u}_{tot}(x, y, z) = \left[\sum_{i=1}^{N_{xt}} \left(\Delta \bar{u}_{lin}\right)_i^2\right]^{1/2}; \quad \left(\Delta \bar{u}_{lin}\right)_i = \sum_{j=1}^{N_r(x_t)} \Delta \bar{u}_j(x, y, z). \tag{3}$$

$N_{xt}$ is the number of unique axial locations where a turbine is located. $N_r(x_t)$ is the number of rotors at the location $x_t$. In this paper, $N_{xt}$ is 1 for the isolated turbine cases, and 5 for the wind farm cases. Furthermore, since we only consider either an

isolated turbine or a wind farm with one column of turbines, $N_r$ is 1 for the 1-rotor cases and 4 for the 4-rotor cases. Finally,

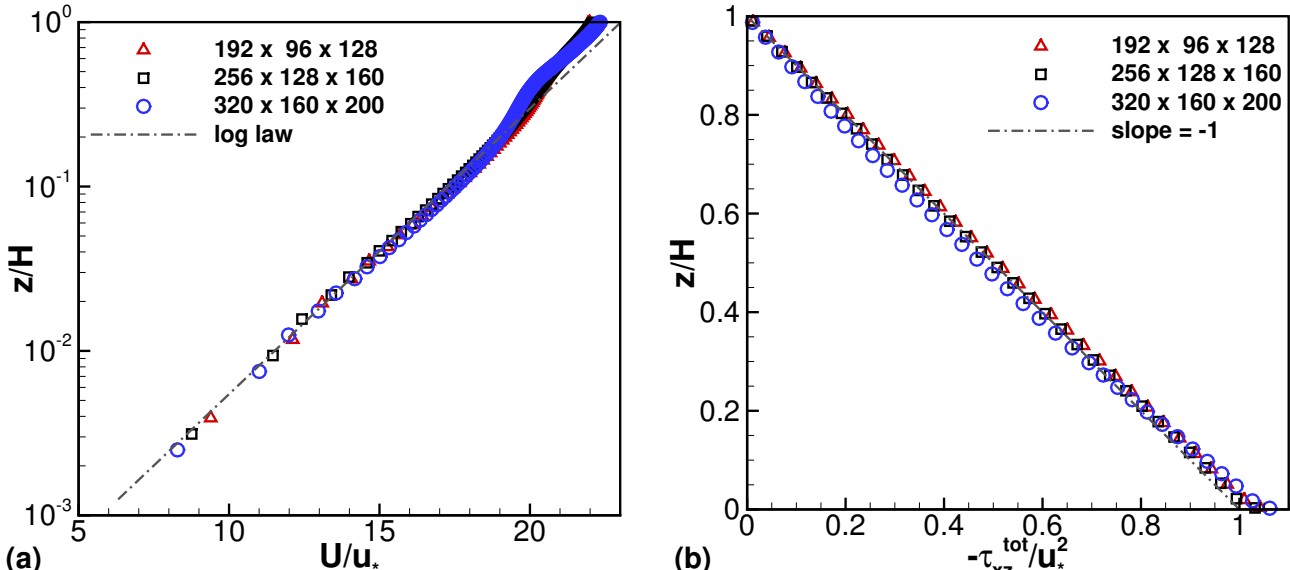

**Figure 2.** Profiles of time- and horizontally-averaged (a) streamwise velocity, and (b) negative of total shear stress from the ABL (precursor) simulations with varying grid sizes. Total shear stress is the sum of resolved, subgrid-scale and wall-modeled components.

the mean velocity at each point in the domain is calculated according to

$$\bar{u}(x,y,z) = \bar{u}_{up}(z) - \Delta\bar{u}_{tot}(x,y,z). \tag{4}$$

The upstream velocity is assumed to follow the logarithmic profile, $\bar{u}_{up}(z) = (u_*/\kappa)\ln(z/z_0)$, with $\kappa = 0.4$.

This modeling framework involves two empirical parameters, $k^*$ and $\sigma_0$. Comments regarding selecting these parameters are provided in the appropriate sections below.

## 3 Isolated Turbine Results

### 3.1 Grid Convergence and Baseline Cases

Precursor ABL simulation results are shown first in Fig. 2. These results are averaged over time and the horizontal directions. As expected, the mean streamwise velocity profiles follow the logarithmic law of the wall, particularly in the lower 20% of the domain. The total shear stress profiles also follow the expected line with slope equal to -1. This indicates that the vertical transport of momentum by the ABL is correctly represented by the numerical method and AMD subgrid-scale model, and that the ABL simulations are statistically stationary. Figure 2 also shows that the spatial resolution employed is adequate for these ABL simulations, since the results are almost independent of the grid size.

Results of an isolated 1-rotor turbine and an isolated 4-rotor turbine with $s/d = 0.05$ are shown in Fig. 3. Vertical profiles in the mid-span planes at several locations downstream of the turbine are shown. The mid-span plane is located at $Y_{cen} = L_Y/2$

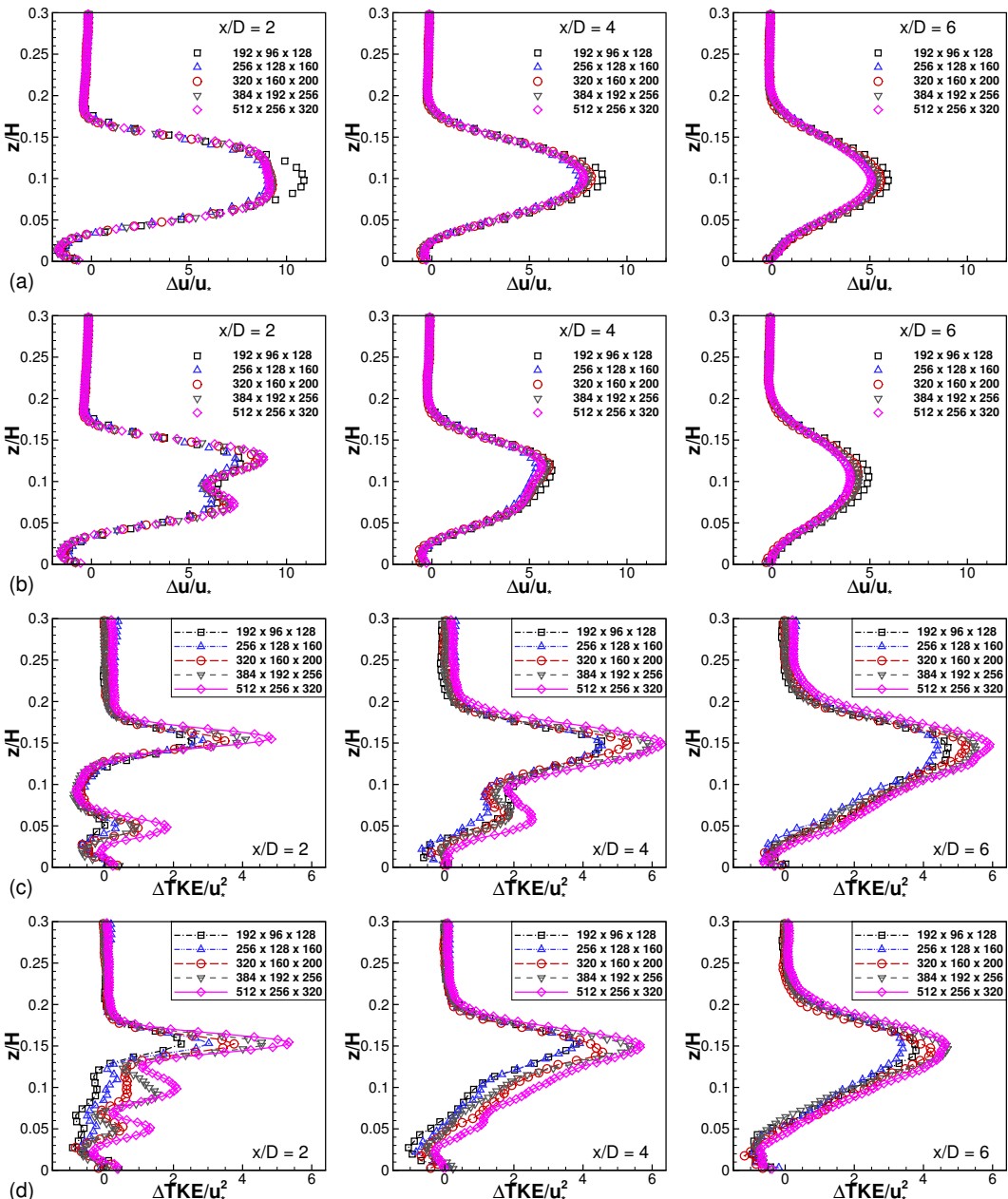

**Figure 3.** Profiles of mean velocity deficit at the centerline and downstream of an isolated (a) 1-rotor turbine and (b) 4-rotor turbine with $s = 0.05d$ for five different grid resolutions. Profiles of added turbulent kinetic energy (TKE) downstream of (c) 1-rotor turbine and (d) 4-rotor turbine with $s = 0.05d$. Mean velocity deficit and added TKE are defined as $\Delta u(x, z) = u(-1D, Y_{cen}, z) - u(x, Y_{cen}, z)$ and $\Delta TKE(x, z) = TKE(x, Y_{cen}, z) - TKE(-1D, Y_{cen}, z)$, respectively. $Y_{cen}$ is $L_Y/2$ for 1-rotor turbine and $L_Y/2 - (1 + s_h)d/2$ for 4-rotor turbine.

for the 1-rotor configuration. The 4-rotor configuration has two mid-span planes, $Y_{cen} = L_Y/2 \pm (1+s_h)d/2$. Results at only one of these, at $L_Y/2 - (1+s_h)d/2$, are shown, since both planes are statistically identical.

Figure 3(a) shows that the velocity deficit profiles for the 1-rotor turbine have a single peak close to $z/H = 0.1$. Two distinct peaks, close to $z/H = 0.1 \pm (1+s_v)d/2$, are seen for the 4-rotor turbine wake in Fig. 3(b) only at $x/D = 2$. Further downstream, at $x/D = 4$ and 6, two distinct peaks are not easily discernible, indicating that the wakes have merged. The added turbulent kinetic energy (TKE) profiles in Figs. 3(c-d) show similar evidence of a single large wake for the 1-rotor turbine and two distinct wakes at $x/D = 2$, which merge further downstream, for the 4-rotor turbine.

Simulations with varying grid sizes (the IT*-s cases) show that the differences between the results reduce as the grid is refined. In general, the sensitivity to grid resolution is larger for the 4-rotor case as compared to for the 1-rotor case. This is expected because the 4-rotor configuration involves smaller length scales, associated with the smaller diameter of the individual rotors, and the tip-spacing. The differences between the velocity deficits obtained using grids G3, G4 and G5 are not easily discernible on the scale of Figures 3(a,b). Differences between the results of grid G2 and those of finer grids are easily apparent only at $x/D = 2$ for the multi-rotor configuration. The double-peaked shape of the velocity deficit at this location is not fully resolved using grid G2, and is better resolved using grids G3 and finer. The velocity deficit values, averaged over the rotor disk regions, for different grid sizes are used to assess grid convergence. Taking the results of grid G5 as reference, the errors in velocity deficits obtained using grid G2 are 3.2% and 1.9% at $x/D = 4$ and $x/D = 6$, respectively.

The added TKE profiles in Figures 3(c,d) show greater sensitivity to grid size than the mean velocity deficits. The resolved portion of the TKE is expected to increase with increasing grid resolution. It should be noted that the resolved TKE cannot be supplemented with a subgrid contribution in an LES using an eddy-viscosity model, where only the deviatoric part of the stress is modeled. Except for a small region close to $z/H = 0.15$ at $x/D = 6$, over most of the domain, the resolved portion of the added TKE is also found to increase with increasing resolution. The turbulence intensity averaged over the rotor area is found to change by around 15% at $x/D = 4$ and 6 between grids G2 and G5. Between grids G2 and G3, the disk-averaged turbulence intensity values vary by 6.5% at $x/D = 4$ and by 3.5% at $x/D = 6$.

A change of 3.2% in the disk-averaged velocity deficit on doubling the grid resolution (from G2 to G5) implies a change of approximately 9.9% in the averaged power. The results pertaining to estimates of power, in particular the comparisons between LES and analytical model predictions, presented in this manuscript should be interpreted keeping this limitation in mind. The computational costs per simulation were approximately 4400 CPU-hours and 70000 CPU-hours on grids G2 and G5, respectively. Even with near-perfect scaling, as was obtained with very careful attention to parallel implementation in our code, in view of the large parameter space to be evaluated, it was decided to conduct all further simulations on grid G2. For the wind farm cases with domain size increased to $1.25\pi$ in the $x$ direction, the number of points in the $x$ direction is increased to 320 to retain the same resolution. This grid is labeled as G2A in Table 1. The grids G2/G2A imply that the smaller rotor disk (diameter $d$) is resolved by $4 \times 8$ points and the composite wake of the multi-rotor turbine (diameter $D$) is resolved by $8 \times 16$ points in the $y - z$ plane. The details in the region between the rotor tips are obviously missed. However, as shown in the next subsection, the overall effect of varying tip spacing is captured, because the actuator disk model appropriately adjusts the distribution of forces across the discretization points. It should be noted that the level of resolution of the composite wake

is in keeping with the recommendation in Wu and Porté-Agel (2011), and is comparable to the grid resolution used by several previous studies (Calaf et al., 2010; Stevens et al., 2014a).

## 3.2    Effect of Tip Spacing

Isolated 4-rotor turbines with varying tip spacings, $s_h = s_v = s$, are studied in this subsection (IT2-s cases). Contours of the mean streamwise velocity deficit and the TKE (Fig. 4) in the mid-span planes show that one large wake immediately downstream of a 1-rotor turbine is replaced by four smaller wakes immediately downstream of the four rotors of the 4-rotor turbines. Comparing Figures 4(a,c,e), it is clear that the wake of a 4-rotor turbine at any downstream location (e.g. at $x/D = 4$), is weaker in magnitude than that of the 1-rotor turbine. This is also seen in the profiles shown in Fig. 5. In other words, the wake of a 4-rotor turbine is seen to recover faster than the wake of a 1-rotor turbine with the same thrust coefficient and rotor area. Figure 5 also shows that greater the tip spacing of the 4-rotor turbine, faster is the wake recovery. This is also indicated by the shortening of the contour lines corresponding to $\Delta u/u_* = 1$ and 2.5 in Fig. 4 with increasing tip spacing.

An intuitive explanation for the increasing rate of wake recovery with increasing tip spacing is as follows. The characteristic length scale of the wake of the 1-rotor turbine is diameter $D$, while that for the individual wakes of the 4-rotor turbines is the smaller diameter $d$. Furthermore, the spacing between the tips of the 4-rotor turbine allows for greater entrainment of low-momentum fluid into the 4-rotor turbine wakes. As a result, the rate of wake recovery is larger for the 4-rotor turbine as compared to the 1-rotor turbine, and increases with increasing $s$.

The wakes of the individual rotors of a 4-rotor turbine expand with downstream distance, and eventually merge to form a single wake. The axial distance where individual wakes of the four rotors may be considered to have merged increases with increasing $s$. This is seen clearly in Fig. 5, where two peaks in the velocity deficit profiles are not seen at $x/D = 4$ for the $s/d = 0.1$ turbine, while two peaks are clearly visible at $x/D = 6$ for the $s/d = 0.5$ turbine.

The contour plot of TKE shown in Fig. 4(b) is strikingly similar to those reported previously (e.g. Fig. 18 in Abkar and Porté-Agel (2015)) for an isolated 1-rotor turbine. The TKE contours in Fig. 4(b) are similar in shape to those in Fig. 4(d) beyond approximately $x/D = 4$, but are quite dissimilar to the contours in Fig. 4(f). This is further evidence for the observation that the wake-merging distance increases with increasing $s$. The rotors of the 4-rotor turbine behave independently up to increasingly larger downstream distances with increasing $s$.

A succinct representation of the effect of tip spacing on the wake of an isolated 4-rotor turbine with respect to that of an isolated 1-rotor turbine is shown in Fig. 6, where rotor-disk-averages of four quantities is plotted as a function of the axial distance . The rotor-disk averages are calculated at each axial ($x/D$) location and over different regions in the $y - z$ plane depending on the turbine configuration. The averages are computed over one disk of diameter $D$, centered at $(L_Y/2, 0.1H)$ for the 1-rotor turbine, and over four disks of diameters $d$ each, centered at $(L_Y/2 \pm (1+s_h)d/2, 0.1H \pm (1+s_v)d/2)$, for the 4-rotor turbines. The disk averaged TI is actually the ratio of the square-root of the disk-averaged TKE and disk-averaged mean streamwise velocity, being slightly different from the disk-average of the point-wise turbulence intensity. The disk-averaged added turbulence intensity is defined as, $\Delta I_{disk}(x) = I_{disk}(x) - I_{disk}(-1D)$, where $I_{disk} = \sqrt{(2/3)}TKE_{disk}$.

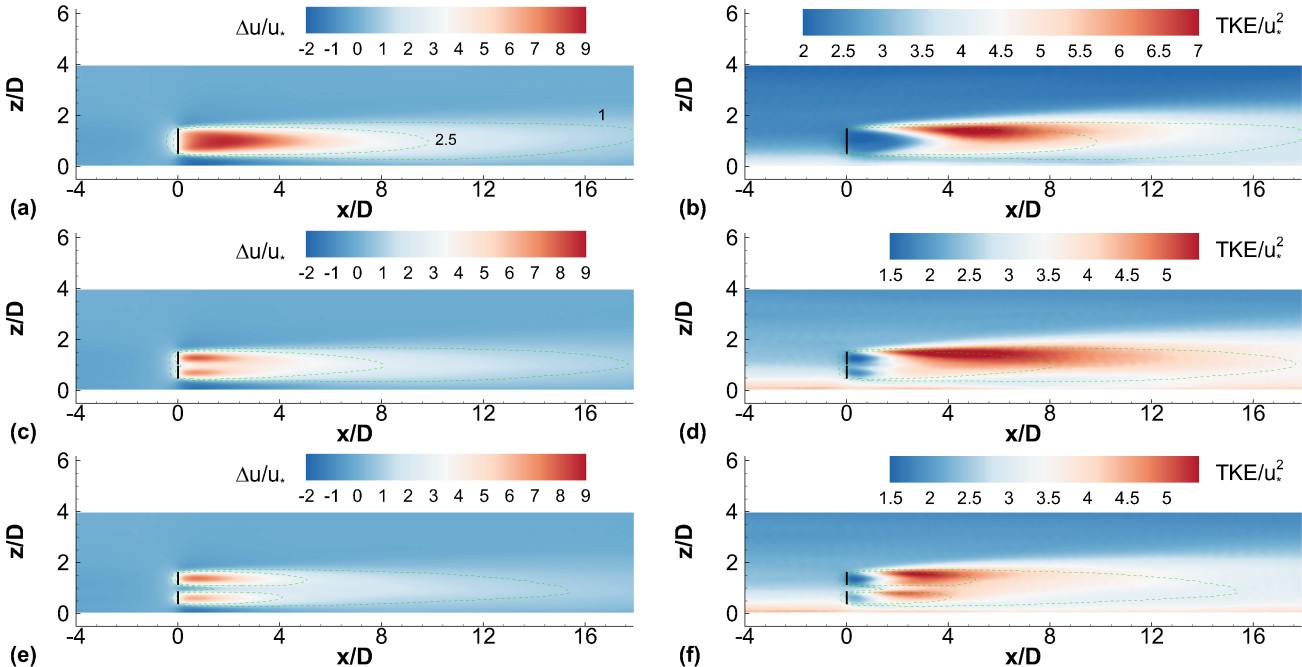

**Figure 4.** Contours of (a,c,e) mean velocity deficit and (b,d,f) TKE at the centerline, for (a,b) 1-rotor turbine, and 4-rotor turbines with tip spacings (c,d) $s = 0.1d$ and (e,f) $s = 0.5d$. Centerline $Y_{cen}$ varies with turbine configuration. Black solid lines denote turbine rotors. Dashed lines are velocity deficit contours corresponding to the levels $\Delta u/u_* = 1$ and 2.5.

Figure 6(a) shows that the streamwise velocity deficits are always smaller for a 4-rotor turbine than for a 1-rotor turbine, and
that deficits decrease monotonically with increasing tip spacing. Interestingly, the 4-rotor turbine with no clearance between the rotor blades (tip spacing $s/d = 0$) also shows reduced velocity deficits in the intermediate downstream region, i.e. $x/D = 4$ and $x/D = 6$. The curves corresponding to the $s/d = 0$ turbine and the $s/d = 1$ turbine act as bounds to the curves corresponding to intermediate tip spacings. The disk-averaged added TKE and $TI_{disk}$ curves (Figures 6(b) and (c), respectively) do not show a monotonic behavior at all downstream locations with increasing $s$. The curves corresponding to the $s/d = 0$ and $s/d = 1$
turbines do not act as bounds for the curves corresponding to the intermediate tip spacings. However, in general, the second order turbulent statistics show a decrease in magnitude with increasing tip spacing.

The disk-averaged added turbulence intensity can be compared to that reported in Figures 18(b,d,f) of van der Laan et al. (2019). For ambient turbulence intensities of 5% and 10% investigated in van der Laan et al. (2019), $\Delta I_{disk}$ values were found to be larger for the 4-rotor case than for the 1-rotor case in the near-wake region, and smaller further downstream. For the largest
ambient turbulence intensity of 20%, the $\Delta I_{disk}$ values for the 4-rotor case were always smaller than for the 1-rotor case. The current LES results are qualitatively similar to the highest ambient turbulence intensity level results in van der Laan et al. (2019), although the ambient turbulence intensity in our current LES is approximately 8%. The reasons for this discrepancy are not clear and should be studied in future work.

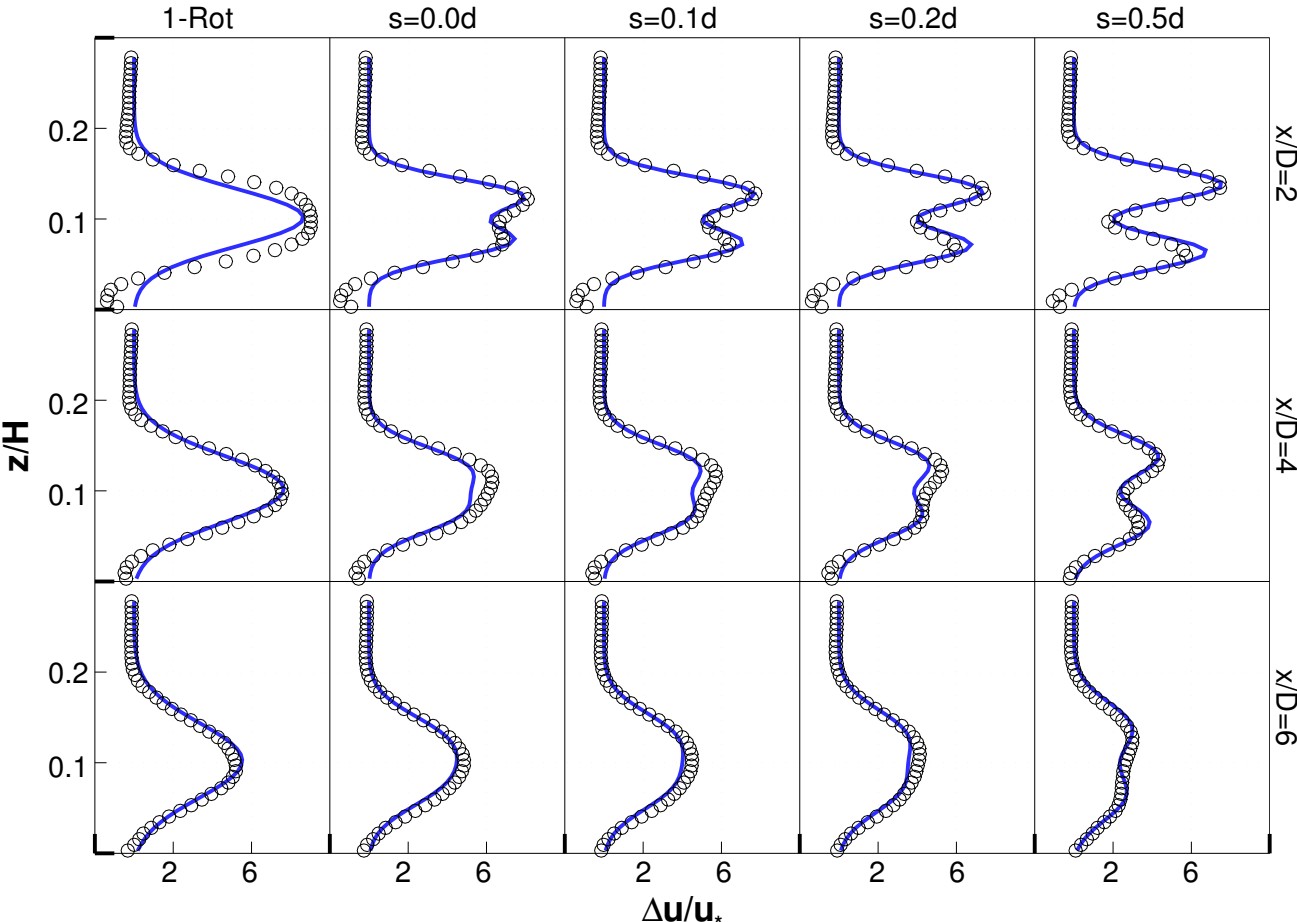

**Figure 5.** Velocity profiles downstream of an isolated 1-rotor turbine and isolated 4-rotor turbines with different tip spacings, $s/d = 0, 0.1, 0.2$ and 0.5. Black symbols are LES results and blue lines are analytical model results.

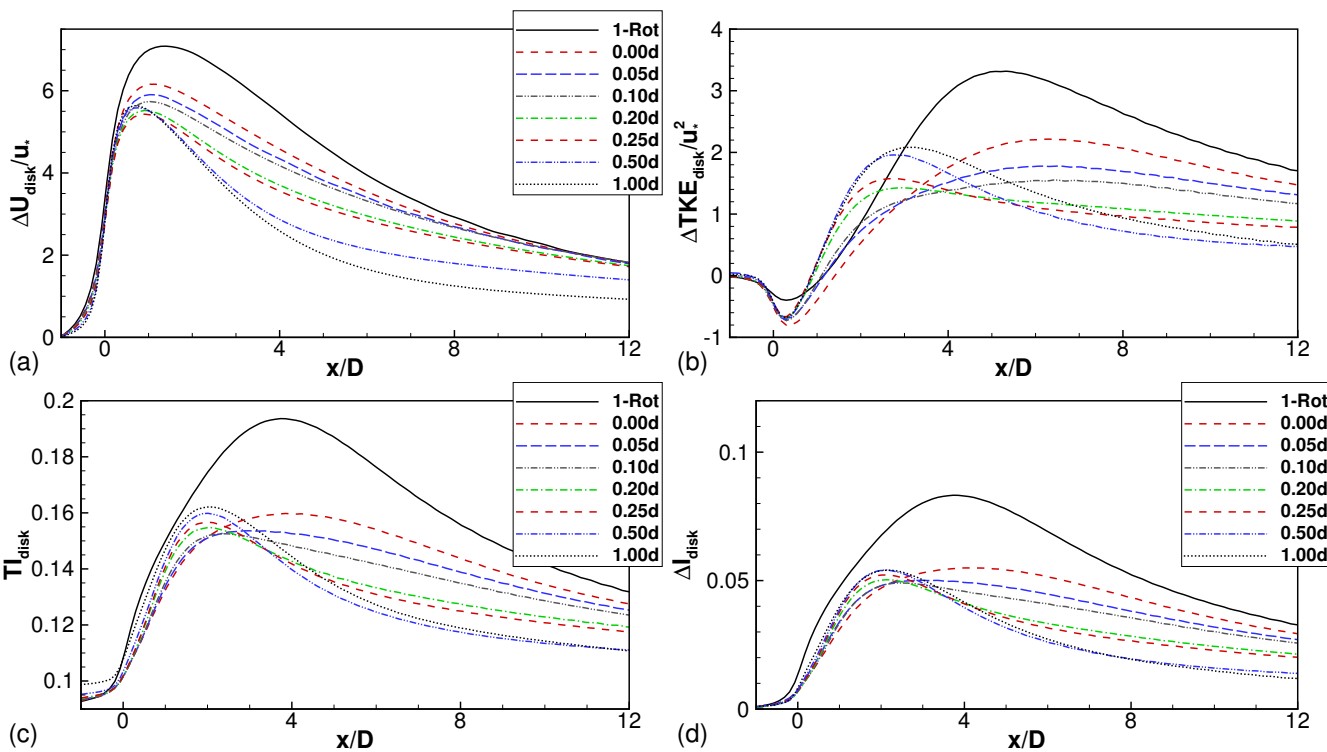

**Figure 6.** Effect of tip spacing on disk-averaged (a) mean velocity deficit, (b) added TKE, (c) turbulence intensity and (d) added turbulence intensity. Disk averages are computed over rotor disk area(s) corresponding to each turbine configuration. Disk-averaged turbulence intensity is the ratio of the square-root of the disk-averaged TKE to the disk-averaged velocity $TI_{disk} = \sqrt{TKE_{disk}}/U_{disk}$. Added disk-averaged turbulence intensity is $\Delta I_{disk} = I_{disk} - I_{disk}(-1D)$, with $I_{disk} = \sqrt{(2/3)}TI_{disk}$.

### 3.3 Effect of Thrust Coefficient

The IT2-$C'_T$ cases, along with two cases from the IT2-s set of simulations, are compared to study the effect of thrust coefficient. Only one 4-rotor configuration, with tip spacing $s/d = 0.1$, is considered here. Figure 7 shows that the trends observed for $C'_T = 4/3$ hold for the other two thrust coefficients studied as well. The disk-averaged velocity deficits are smaller for the 4-rotor turbine than for the corresponding 1-rotor turbine. The added TKE (not shown) and $TI_{disk}$ are also smaller for the 4-rotor turbine than for the 1-rotor turbine for all the thrust coefficients studied.

### 3.4 Analytical Model

The analytical modeling framework predicts the mean velocity deficits of the 1-rotor and 4-rotor turbines accurately. Empirical parameters values $k_* = 0.025$ and $\sigma_0/d_0 = 0.28$ were found to lead to accurate predictions for all the cases investigated. Here, $d_0$ equals $D$ for the 1-rotor cases and equals $d$ for the 4-rotor cases. These values of $k_*$ and $\sigma_0$ are slightly different from those proposed in Bastankhah and Porté-Agel (2016), but within the range mentioned in Bastankhah and Porté-Agel (2014).

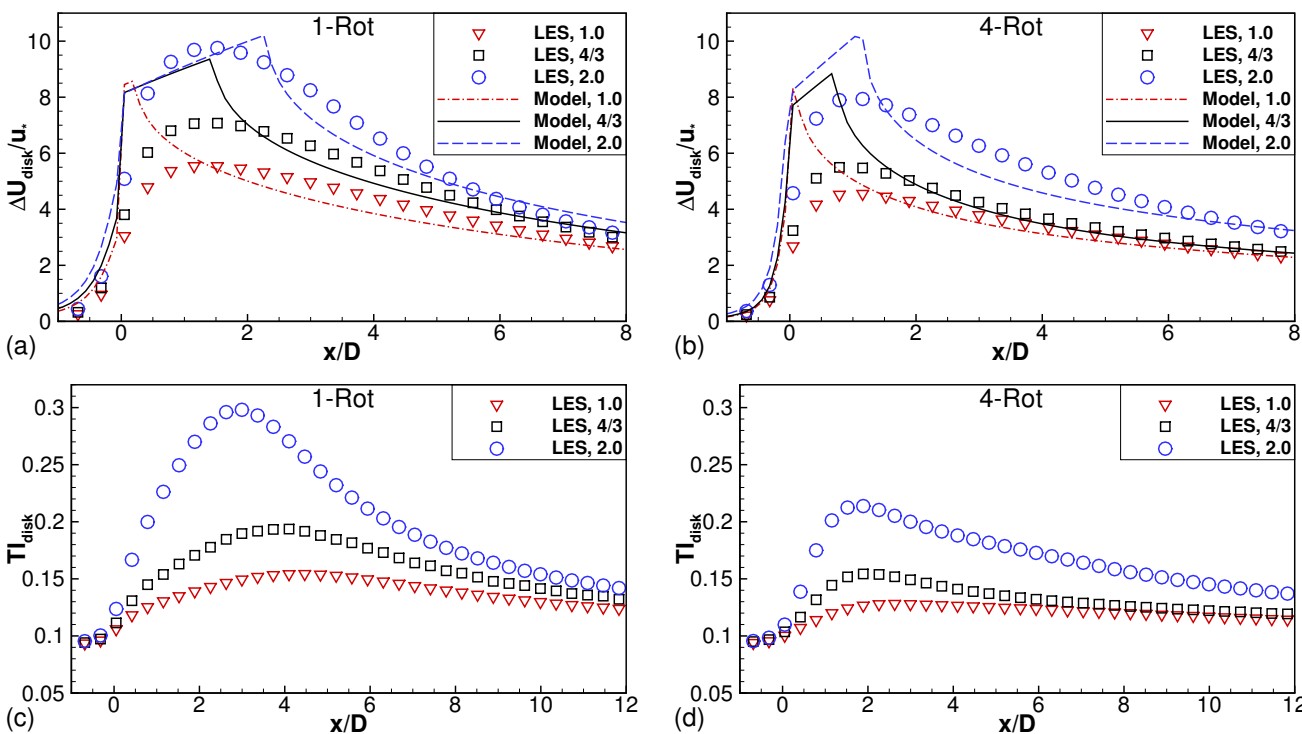

**Figure 7.** Effect of thrust coefficient on disk-averaged (a,b) velocity deficit and (c,d) turbulence intensity for (a,c) 1-rotor turbine and (b,d) 4-rotor turbine with $s/d = 0.1$.

In particular, Fig. 5 shows that the radial profiles of the velocity deficit at several downstream locations, and for turbines with different tip spacings, are predicted quite accurately. Slight under-predictions or over-predictions are observed very close to the turbine, but the overall predictions are accurate, particularly beyond $x/D = 2$. Disk-averaged velocity deficit profiles are also predicted accurately, but are not shown on Fig. 6(a) to avoid clutter. Figures 7(a-b) show that the Gaussian analytical model is reasonably accurate at predicting the disk-averaged velocity deficit for all thrust coefficients beyond the very-near-wake region, i.e. approximately beyond $x/D = 2$.

## 4   Multi-Turbine Simulation Results

Wind farms comprised of a line of five turbines aligned with each other and with the mean wind direction are studied here. These cases are labeled WF* in Table 1.

## 4.1 Effect of Tip Spacing

The effect of tip spacing on the contours of velocity deficit and TKE are seen in Fig. 8. The axial spacing between different turbines in the wind farm is kept fixed at $4D$ and the thrust coefficient is $4/3$ for all rotors of all turbines. It is clear that the velocity deficits are significantly different between the 1-rotor and 4-rotor wind farms, as well as between 4-rotor wind farms with different tip spacings. The single wake behind the turbines in the 1-rotor wind farm are replaced by four smaller wakes behind the turbines in the 4-rotor wind farms. The wakes move further apart in the radial directions as the tip spacing increases. Similar to the TKE distribution behind an isolated 1-rotor turbine, the TKE values are largest around the top-tip height of the turbines.

The effect of tip spacing on 4-rotor wind farms is quantified in Fig. 9. Focusing on Fig. 9(a-b), the profiles of the velocity deficits averaged over the rotor disk and $TI_{disk}$ have local maxima close to the turbine locations, i.e. at $x/D = 0$, 4, 8, 12 and 16. The velocity deficit profile for the 1-rotor wind farm has a maximum close to turbine 2 (located at $x/D = 4$), as seen in Figures 9(a) and 8(a). The velocity deficit profile saturates from turbine 3 onward, i.e. the local maxima at $x/D = 8, 12$ and 16 have approximately equal magnitudes. The $TI_{disk}$ profiles in Fig. 9(b) show similar behavior for the 1-rotor wind farm.

The velocity deficits of the 4-rotor turbines are seen in Fig. 9(a) to be smaller than those of the 1-rotor turbine for the first two turbines ($x/D = 0, 4$). In this region, $x/D < 8$, the deficits decrease with increasing tip spacing, which is consistent with the observations for isolated turbines (Fig. 6(a)). The deficits accumulate and the disk-averaged profiles for all 4-rotor wind farms are almost equal to that for the 1-rotor wind farm for turbine rows 3 onward (for $x/D > 8$). The turbulent intensity profiles are smaller for the 4-rotor wind farms than for the 1-rotor wind farm, and decrease with increasing $s/d$. This sensitivity to the tip spacing persists downstream of all turbines, unlike the velocity deficits, which are sensitive only downstream of the first two turbines.

The relative powers of the turbines are shown in Fig. 9(c). The power of the first (or front) turbine is used for normalization in each wind farm. Thus, the relative power for turbine $i$ is calculated as $P_i/P_1 = \overline{u_i^3}/\overline{u_1^3}$, where the overhead bar represents time-averaging and subscript $i$ denotes the location of the turbine within the wind farm. The relative power of turbine 2 ($x/D = 4$) in the 1-rotor wind farm is minimum, and the relative power profile shows a slight recovery for turbines 3-5. This is consistent with the maximum for the velocity deficit at turbine 2, seen in Fig. 9(a). The relative powers of turbines in the 4-rotor wind farms are sensitive to the tip spacing as well as the turbine location. For $s/d = 0.1$, only turbine 2 has larger relative power than turbine 2 of the 1-rotor wind farm, while for $s/d = 0.5$, turbines 2-4 have larger relative powers than the corresponding turbines of the 1-rotor wind farm. All these trends are consistent with the velocity deficit profiles seen in Fig. 9(a). These results are consistent with the findings of van der Laan and Abkar (2019), where the benefit was restricted to only the first downstream turbine row for tip spacing of 0.1d. Our results further quantify how far downstream into the wind farm the benefit propagates with increasing tip spacing.

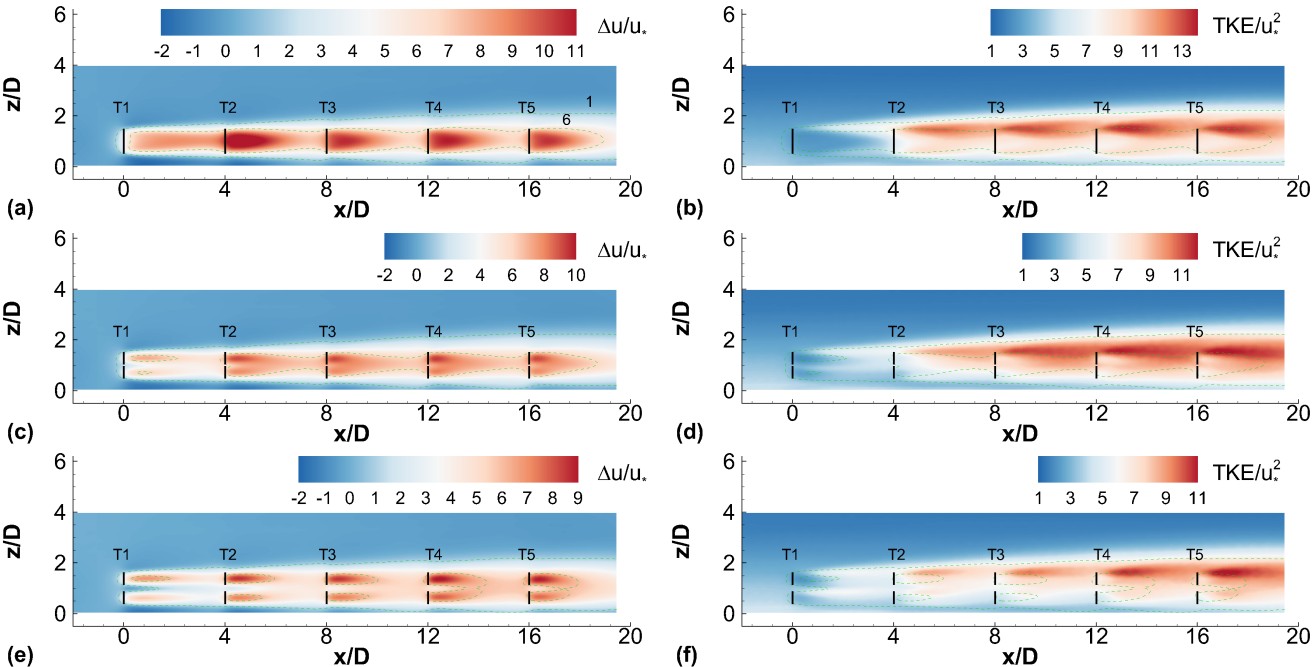

**Figure 8.** Contours of (a,c,e) streamwise velocity deficit and (b,d,f) TKE at the centerline for (a,b) 1-rotor wind farm, and 4-rotor wind farms with tip spacings (c,d) $s/d = 0.1$ and (e,f) $s/d = 0.5$. Axial spacing is $4D$ in each wind farm. Black lines denote turbine rotors. Dashed lines are velocity deficit contours corresponding to levels $\Delta u/u_* = 1$ and 6.

## 4.2 Effect of Axial Spacing and Thrust Coefficient

The effect of axial spacing on the performance of 4-rotor wind farms can be studied by comparing Figures 9(d-f) to Figures 9(a-c). While the same qualitative trends are seen for axial spacings of $S_X = 4D$ and $6D$, there are significant quantitative differences. The larger spacing between turbines in the $6D$ wind farms allows the wakes to recover to a greater extent before another turbine is encountered. Thus, the disk averaged velocity deficits and turbulence intensities are, in general, smaller in the wind farms with axial spacing of $6D$. Consequently, comparing Figures 9(c) and (f), the relative power values are larger for wind farms with larger axial spacing.

Interaction between the effects of tip spacing and axial spacing are also seen on comparing Figures 9(c) and (f). For instance, the relative powers of turbines 2 and 3 of the wind farm with $s/d = 0.5$ are appreciably larger than the corresponding turbines of the 1-rotor wind farm, when the axial spacing is $4D$. However, relative power of only turbine 2 of the wind farm with tip spacing $s/d = 0.5$ is appreciably larger than that of the corresponding 1-rotor wind turbine, when the axial spacing is increased to $6D$. Thus, tip spacing has a greater effect on the relative power in a closely spaced wind farm.

Figure 10 shows that the trends observed for $C_T' = 4/3$ hold for other values of thrust coefficient as well. The velocity deficit and turbulence intensity are larger for cases with larger thrust coefficient. For each value of $C_T'$, the velocity deficit of the 4-

rotor wind farm is generally smaller than that of the 1-rotor wind farm downstream of the first two turbines (for approximately $x/D < 8$) and are almost equal beyon this. Since the tip spacing of the 4-rotor wind farm is $s/d = 0.1$, only turbine 2 shows a larger relative power in the 4-rotor wind farm compared to the 1-rotor wind farm, consistent with the observation made in Figure 9. For $C'_T = 2$, the velocity deficit profiles cross over, and the 4-rotor profile is larger than the 1-rotor profile, in a small region upstream of turbine 3. As a result, the relative power of turbine 3 is smaller in the 4-rotor wind farm compared to the 1-rotor wind farm. However, this crossover in power is smaller in magnitude than the values for turbine 2, such that the collective relative power of the downstream turbines is larger for the 4-rotor wind farm than for the 1-rotor wind farm.

The effect of all governing parameters $(s, S_X, C'_T)$ on the wake losses in multi-rotor wind farms is presented succinctly in Fig. 11. Figure 11(a) shows the average power of turbines 2 through 5 ($P_{2-5} = (1/4) \sum_{i=2}^{5} P_i$), normalized by the power of the front turbine in each wind farm. Aggregation of relative powers across all downstream rows, as done here, can hide negative power differences (associated with the crossovers referred to above) that might occur at individual turbine rows. Despite this, the aggregated relative power is a useful measure of the overall wake losses associated with a particular wind farm. It is seen that $P_{2-5}/P_1$ is larger for all 4-rotor wind farms than the corresponding 1-rotor wind farm with the same thrust coefficient and axial spacing. The benefit increases with increasing tip spacing.

Each data point in Fig. 11(a) is normalized by the power of the front turbine in the respective wind farm. The front turbine power is expected to be similar to that of an isolated turbine, and hence, is expected to be dependent on the thrust coefficient, but not on the axial spacing. This is seen to be the case in Fig. 11(b), where the power of the front turbine extracted from the different wind farm cases are shown. For comparison across cases with different thrust coefficients, all powers are normalized by the power of the front turbine in the 1-rotor wind farm with the same thrust coefficient. The front turbine powers are independent of the axial spacing, and lines corresponding to $S_X = 5D$ and $6D$ lie on top of the line corresponding to $S_X = 4D$. Figure 11(b) also shows that the front turbine power in 4-rotor wind farms is weakly dependent on the tip spacing. As the tip spacing varies over $s/d = 0.1$ to $0.5$, the front turbine power varies by $3.5\%$, $2.7\%$ and $3.2\%$, with the thrust coefficients fixed at $1, 4/3$ and $2$, respectively. We note that this variation cannot be explained by the variation in power potential due to different tip spacings (see Appendix B), and is likely caused by the effects of turbulent mixing in the wake (Nishino and Wilden, 2012), which are different for different tip spacings.

To account for the differences in the front turbine power, the average power of turbines 2 through 5 is replotted in Fig. 11(c), with only the 1-rotor front turbine powers used for normalization. The same qualitative conclusions can be drawn from Fig. 11(c), as were drawn from Fig. 11(a), although the magnitudes of the benefit are larger. Finally, the differences between the relative powers of the 4-rotor and 1-rotor configurations are plotted in Fig. 11(d). This plot is directly derived from Fig. 11(c) by subtracting the data points corresponding to the 1-rotor wind farm from the 4-rotor wind farm data, i.e. $\Delta P_{2-5} = P_{2-5} - P_{2-5}^{1-Rot}$. This quantity measures the extent by which wake losses in a 4-rotor wind farm are smaller than wake losses in a 1-rotor wind farm with the same inter-turbine spacing and with all rotors operating with the same thrust coefficient. The benefit of 4-rotor wind farms increases with increasing tip spacing and with decreasing thrust coefficient. The effect of axial spacing on the benefit is slightly ambiguous. For a fixed thrust coefficient and tip spacing, the benefits are largest for $S_X = 4D$, and are almost equal for $S_X = 5D$ and $6D$.

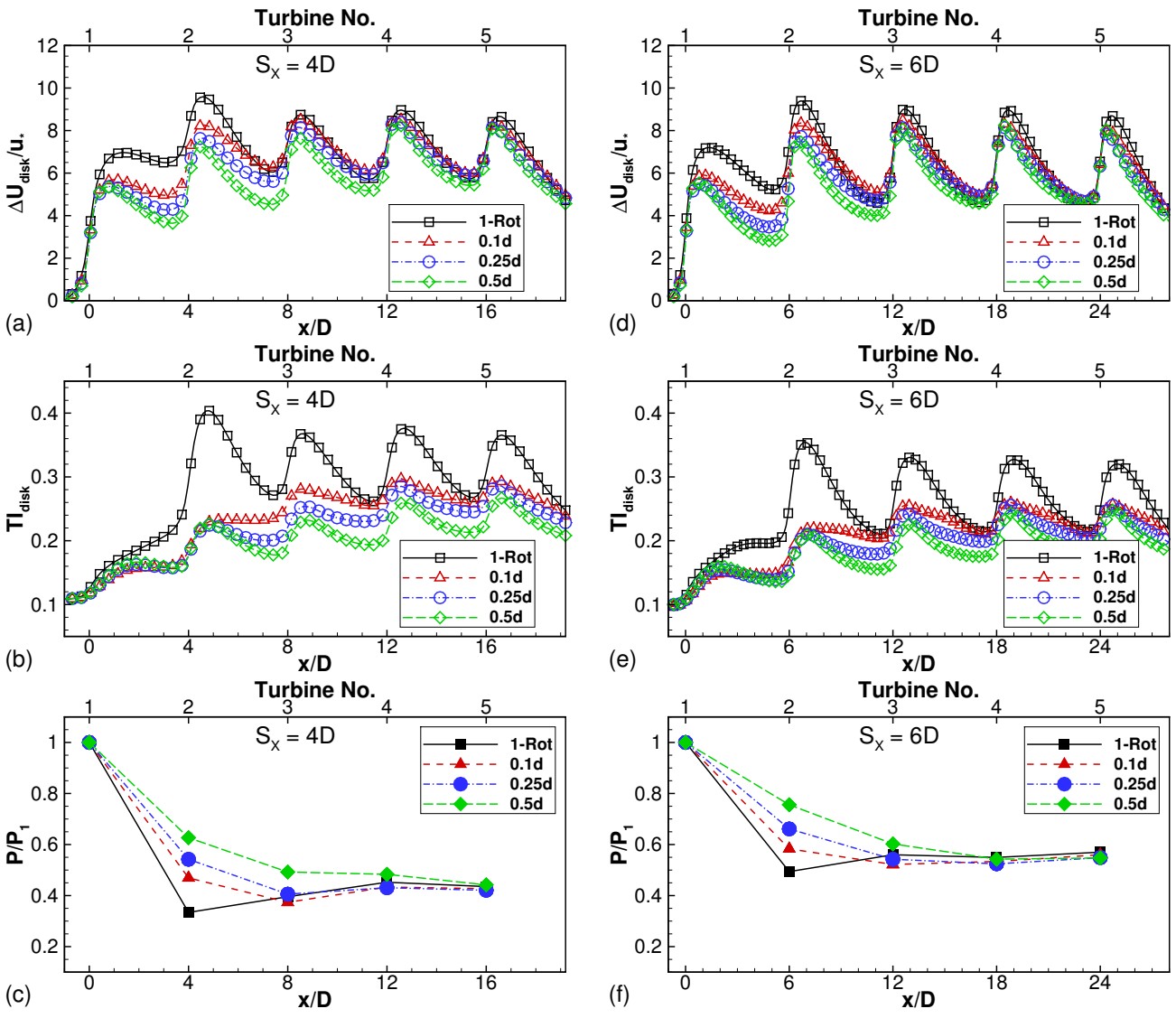

**Figure 9.** Disk-averaged (a) velocity deficit and (b) turbulence intensity, and (c) relative power for wind farms with $S_X = 4D$ and varying tip spacings. Power is normalized by front turbine in each wind farm to compute relative power. (d,e,f) Corresponding results for wind farms with $S_X = 6D$.

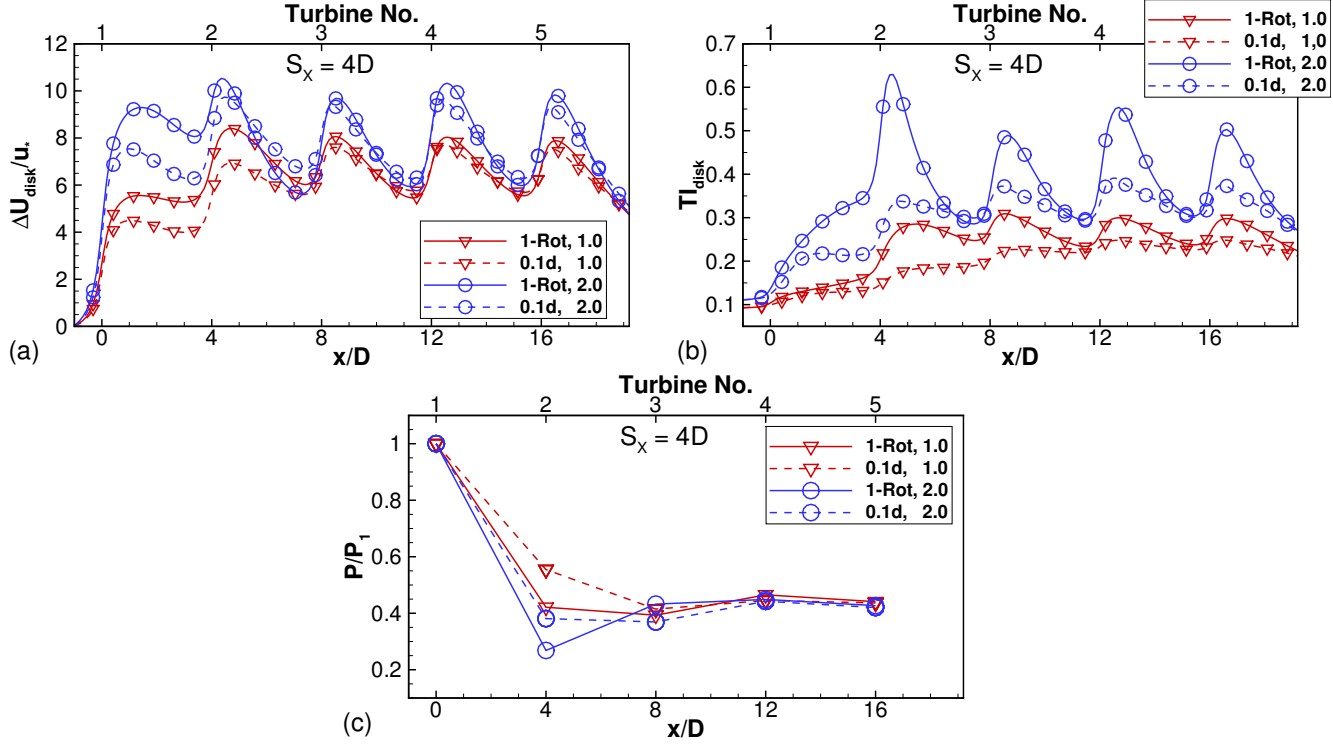

**Figure 10.** Disk-averaged (a) velocity deficit and (b) turbulence intensity, and (c) relative power for 1-rotor and 4-rotor wind farms with $S_X = 4D$ and varying thrust coefficient. Legend denotes the pair $(s/d, C'_T)$.

Appendix C shows that the conclusions drawn above are not affected by the fact that the first turbine powers are significantly different between the 1-rotor and 4-rotor wind farms.

### 4.3 Analytical Model

Predictions of the analytical modeling framework for wind farms comprised of a line of five turbines are examined in this section. The parameter $k_*$, which controls the growth rate of the wake, is extracted from all the 1-rotor wind farm LES. First, the wake widths in the $y$ and $z$ directions are calculated using the definition outlined in Bastankhah and Porté-Agel (2016).

$$\sigma_y(x) = \frac{1}{\sqrt{2\pi}\Delta\bar{u}_{max}(x)} \int_{-\infty}^{\infty} \Delta\bar{u}(x, \hat{y}, Z_{cen})\, d\hat{y}, \tag{5}$$

$$\sigma_z(x) = \frac{1}{\sqrt{2\pi}\Delta\bar{u}_{max}(x)} \int_{-\infty}^{\infty} \Delta\bar{u}(x, Y_{cen}, \hat{z})\, d\hat{z}, \tag{6}$$

where $(Y_{cen}, Z_{cen}) = (L_Y/2, 0.1H)$ are the mid-span and mid-vertical planes of the 1-rotor wind turbine wakes, and $\Delta\bar{u}_{max}(x)$ is the maximum of the velocity deficit at location $x$. The wake width is then calculated as the geometric mean of the wake widths in the two transverse directions, $\sigma = \sqrt{\sigma_y \sigma_z}$.

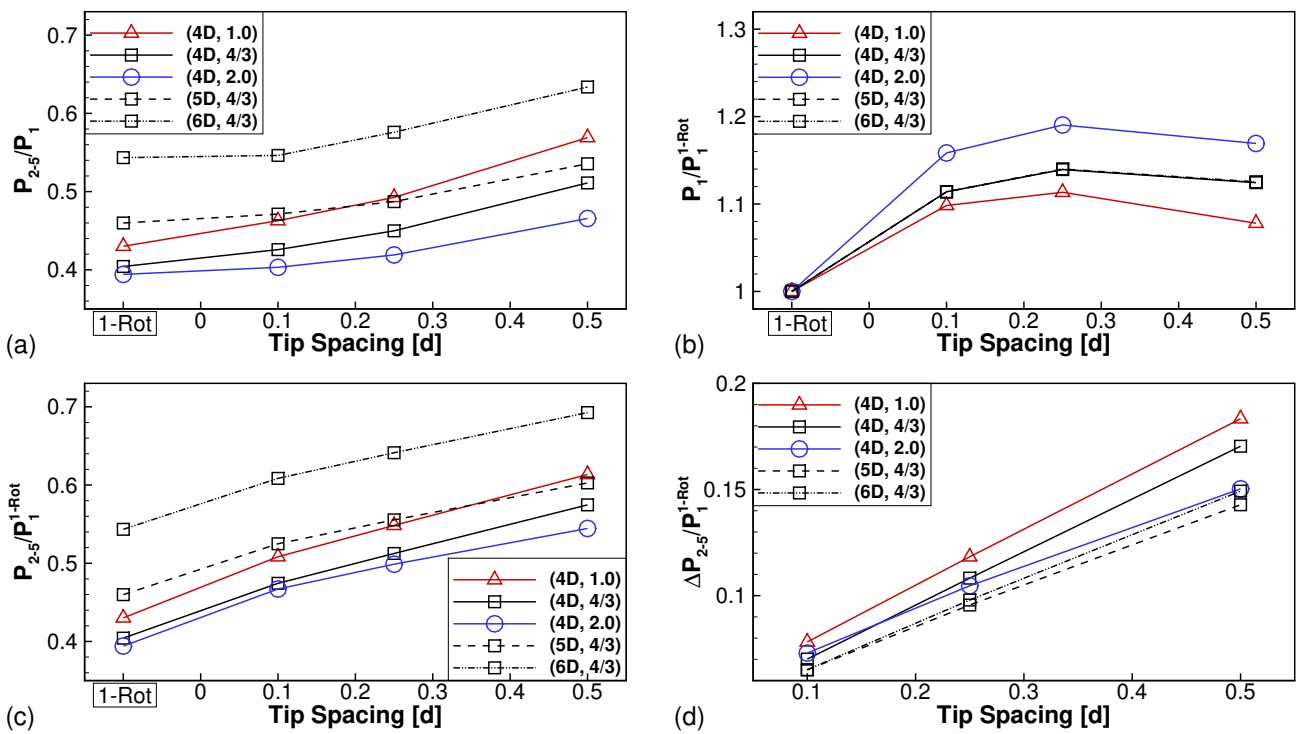

**Figure 11.** Effect of tip spacing, thrust coefficient and axial spacing on (a) power of turbines 2 through 5 normalized by power of front turbine, (b) power of front turbine and (c) power of turbines 2 through 5 normalized by power of front turbine of corresponding 1-rotor wind farm. (d) Benefit of 4-rotor wind farms over corresponding 1-rotor wind farm.

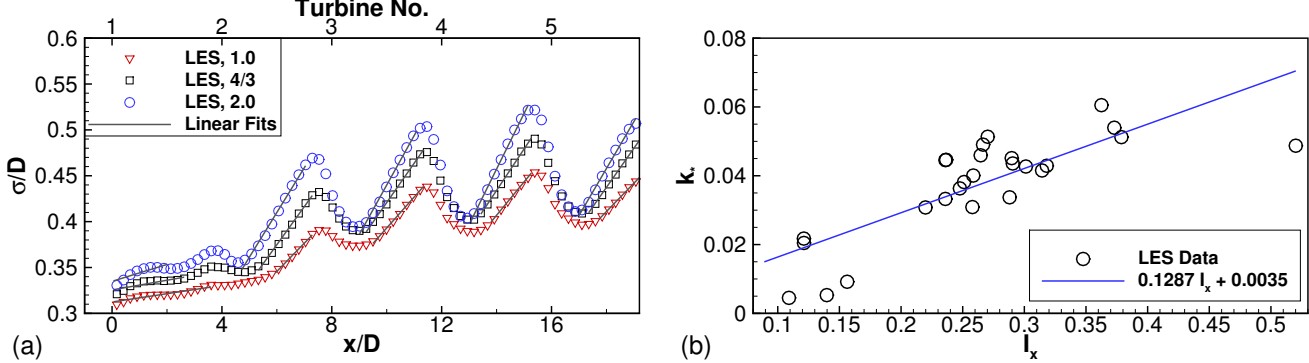

**Figure 12.** (a) Wake width, $\sigma/D$, extracted from LES of 1-rotor wind farms with axial spacing $S_X = 4D$ and varying thrust coefficient indicated in the legend. Slopes of black fitting lines give wake growth rate parameter $k_*$. (b) Wake growth rate parameter as a function of disk-averaged streamwise turbulence intensity extracted from all LES of 1-rotor wind farms. Blue line is the linear fit to the LES data.

Wake widths extracted from three 1-rotor LES with fixed $S_X = 4D$ and varying thrust coefficient are shown in Fig. 12(a). Turbines are located at $x/D = 0, 4, 8, 12$ and 16 in this plot. Moving downstream from one turbine location, the wake widths generally increase, until the effect of the next downstream turbine is felt. The wake width profiles show dips close to the turbine locations, followed by regions of growth. Regions where the wake widths grow approximately linearly are identified with black solid lines in Fig. 12(a). These black solid lines are linear fits to the data, and the extents of the linear fitting region are identified visually. The slopes of these lines yield the wake growth rate parameter, $k_*$.

The wake growth rate parameter values for all turbines in the 1-rotor wind farm simulations are compiled in Fig. 12(b). The $k_*$ values are plotted against the streamwise turbulence intensity, $I_x$, at each turbine rotor disk. As observed in previous studies, the wake growth rate increases with increasing turbulence intensity. The solid blue line fits the data with a correlation coefficient of 0.8. In subsequent model runs for 1-rotor and 4-rotor wind farms, this linear regression model is used to determine $k_*$, with $I_x$ extracted from the LES results.

Model predictions are compared to LES results for two cases in Fig. 13. The sensitivity of the model predictions to the second tunable parameter, the initial wake width $\sigma_0$, is seen in this figure. Figure 13(a) shows that the disk-averaged velocity deficit is over-predicted by the analytical model with $\sigma_0/D =$ very close to the turbines, while it is under-predicted (to a lesser degree) with $\sigma_0/D = 0.32$. Farther away from the turbines, approximately between $1D$ to $3D$ downstream of each turbine, using $\sigma_0/D = 0.28$ yields good agreement with the LES results, while using $\sigma_0/D = 0.32$ continues to yield under-predictions. The power predictions shown in Fig. 13(b) also show sensitivity to the value of $\sigma_0$. The relative power of turbine 2 is captured accurately with $\sigma_0/D = 0.28$, while the relative powers of further downstream turbines are under-predicted by around $10\%$. With $\sigma_0/D = 0.32$, the relative power of turbine 2 is over-predicted, while that of further downstream turbines is in better agreement with the LES results. Similar conclusions can be drawn from the results of the 4-rotor turbine with $s/d = 0.1$, shown in Figures 13(c) and (d). In summary, $\sigma_0/D = 0.28$ leads to better prediction of the mean velocity deficit in the wake region ($1D - 3D$ downstream), while $\sigma_0/D = 0.32$ leads to better prediction at the turbine locations, as evidenced by the better predictions of the power. Thus, the combination of model parameters which leads to accurate predictions in the wake does not necessarily lead to accurate predictions of power, for which, the values at and very close to the turbines need to be predicted accurately.

The influence of using spatially constant values for the wake growth rate parameter on the model predictions is shown in Figure 14. Predictions for two values of $k_*$ (0.025 and 0.04) are shown for each of the two values of $\sigma_0/D$. Predictions for intermediate values of $k_*$ are not shown but lie within the bounds shown by the lines corresponding to $k_* = 0.025$ and 0.04. It is seen that using a spatially non-varying $k_*$ leads to a gradual decrease in the relative power with turbine number. The LES results show the characteristic feature of recovery of the relative power after turbine 2 in the 1-rotor wind farm and after turbine 3 in the 4-rotor wind farm. This feature is not captured for any combination of $\sigma_0/D$ and non-varying $k_*$. Comparing Figures 14(a,b) and Figures 13(b,d) respectively, it is clear that the power degradation recovery is better captured using $k_*$ that varies spatially depending on the local turbulence intensity. Similar observations were reported previously for 1-rotor wind farms (Niayifar and Porté-Agel, 2016), and are seen here to hold for several 4-rotor wind farms as well. It is possible for some cases, particularly the $s/d = 0.5$ wind farms, where the relative power continues to gradually decrease until the fifth turbine

(see Figures 15 and 16), to be better predicted using a spatially constant $k_*$ value. However, no single combination of spatially-constant $k_*$ and $\sigma_0/D$ values was found that resulted in good predictions for all cases. In view of the cases investigated here, we prefer the use of a spatially-varying $k_*$ dependent on the local turbulence intensity, consistent with previous studies for 1-rotor wind farms (Niayifar and Porté-Agel, 2016).

Relative power predictions for all the wind farm cases are compared to LES results in Figures 15 and 16. The average error
in predicting the relative powers of turbines 2 through 5 are shown in each case. The $k_*$ values are obtained as outlined above, while $\sigma_0/d_0 = 0.28$ is used for all cases, where $d_0$ equals $D$ for the 1-rotor cases and equals $d$ for the 4-rotor cases. The absolute errors in relative power averaged over turbines 2 through 5 $((1/4)\sum_{i=2}^{5} |(P_i/P_1)^{LES} - (P_i/P_1)^{model}|)$ are shown in Figures 15 and 16. It should be noted that this level of accuracy is similar to that observed in previous studies (Stevens et al., 2015, 2016) of wind farms that are finite in axial as well as spanwise directions, and where the wind is directed along only one
direction, or averaged over a very narrow (less than $2°$) sector.

The errors are seen to be smallest for the 1-rotor cases. For 1-rotor wind farms, typically, the power of the second turbine is smallest, and there is a slight recovery for turbines 3, 4 and 5. This behavior is reproduced well by the analytical model. In the 4-rotor cases, the relative power saturates farther into the wind farm, typically at the third row for $s/d = 0.1$ and 0.25. For $s/d = 0.5$, the power continues to decrease until the fifth row for most cases. The model predictions, on the other hand,
typically saturate by the second row. Thus, the errors are largest for the second row, although the relative power level of turbines in the fourth and fifth rows is typically well captured.

In conclusion, the analytical modeling framework is capable of reproducing LES results of 1-rotor and 4-rotor wind farms with reasonable accuracy, comparable to previous results for 1-rotor turbines (Stevens et al., 2015). Improved prediction of the region very close to the turbine is needed to further improve the accuracy of the model at predicting the power degradation and
415 wake losses in wind farms.

## 5   Discussion and Summary

This paper is devoted to studying the turbulent wake of a multi-rotor wind turbine configuration, and to comparing it with a conventional single-rotor wind turbine wake. The potential benefits offered by this configuration, with four rotors (with diameters $d = D/2$) mounted on a single tower, over the conventional single-rotor turbine (with diameter $D$) are studied in
detail. Large eddy simulation is used as the primary tool for this work, Applicability of an analytical modeling framework based on the assumption of Gaussian radial profiles of velocity deficits to the multi-rotor configuration is also examined.

The LES results outlined in Sect. 3 show that an isolated 4-rotor turbine wake recovers faster compared to an isolated 1-rotor turbine wake. The isolated 4-rotor turbine wake also shows smaller TKE levels in the rotor disk region. A simple physical reason for this faster wake recovery and lower TKE levels is that the greater perimeter-to-area ratio of the multi-rotor turbine
allows for greater entrainment of low momentum fluid into the wake. The behavior of the wake is sensitive to the tip spacing $(s/d)$, with faster wake recovery seen for for larger $s/d$. This is consistent with the simple physical reasoning presented above, since if $s/d$ is very large, each rotor of the multi-rotor turbine behaves independently of other rotors, and the wake of each

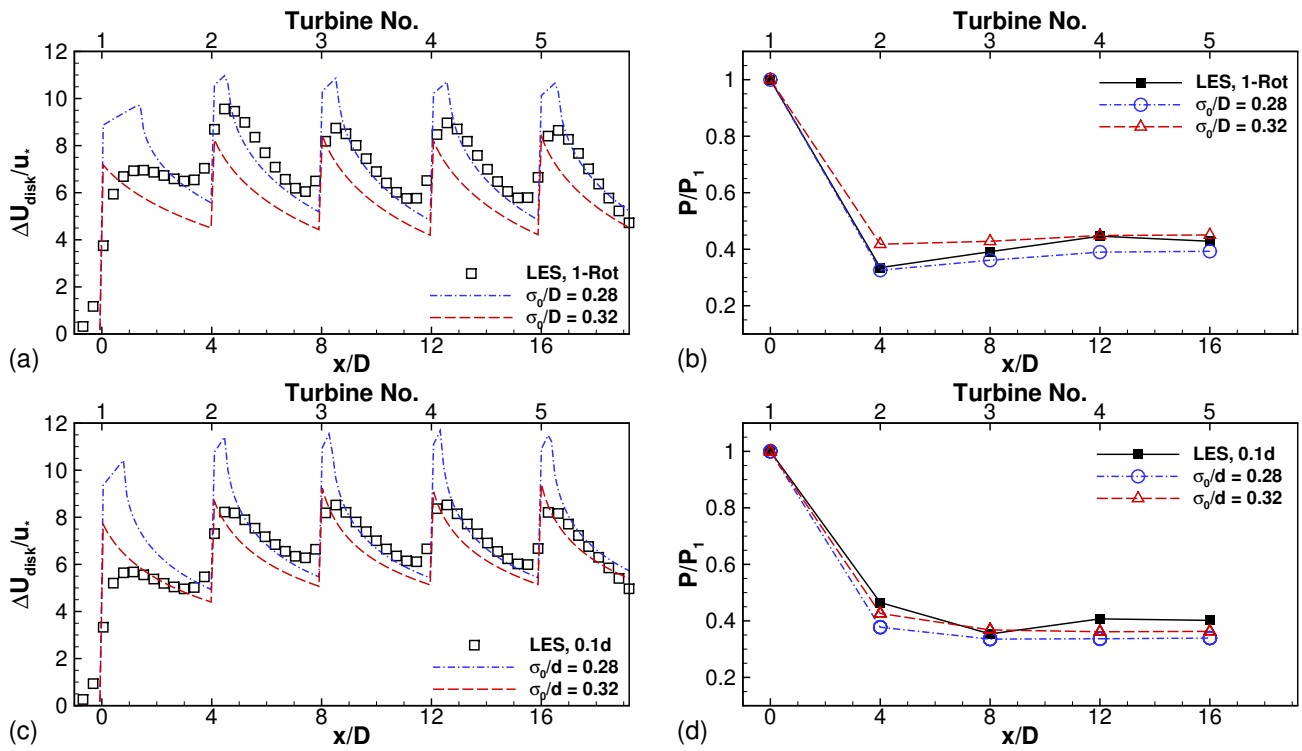

**Figure 13.** LES results and model predictions of (a) disk-averaged velocity deficit and (b) relative power for 1-rotor wind farm with $S_X = 4D$ and $C_T' = 4/3$. (c,d) Corresponding results for 4-rotor wind farm with $s = 0.1d$.

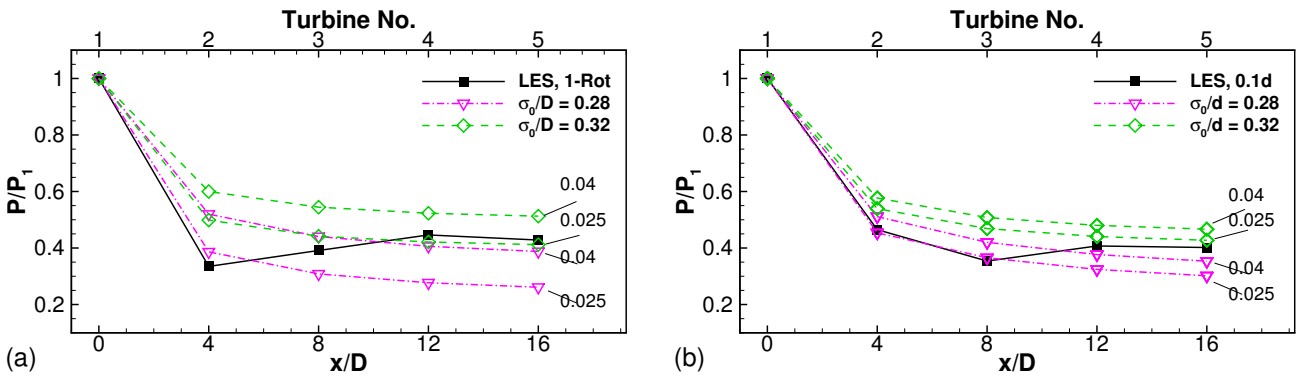

**Figure 14.** LES results and model predictions of relative power using spatially constant $k_*$ for (a) 1-rotor wind farm and (b) 4-rotor $s/d = 0.1$ wind farm with $S_X = 4D$ and $C_T' = 4/3$.

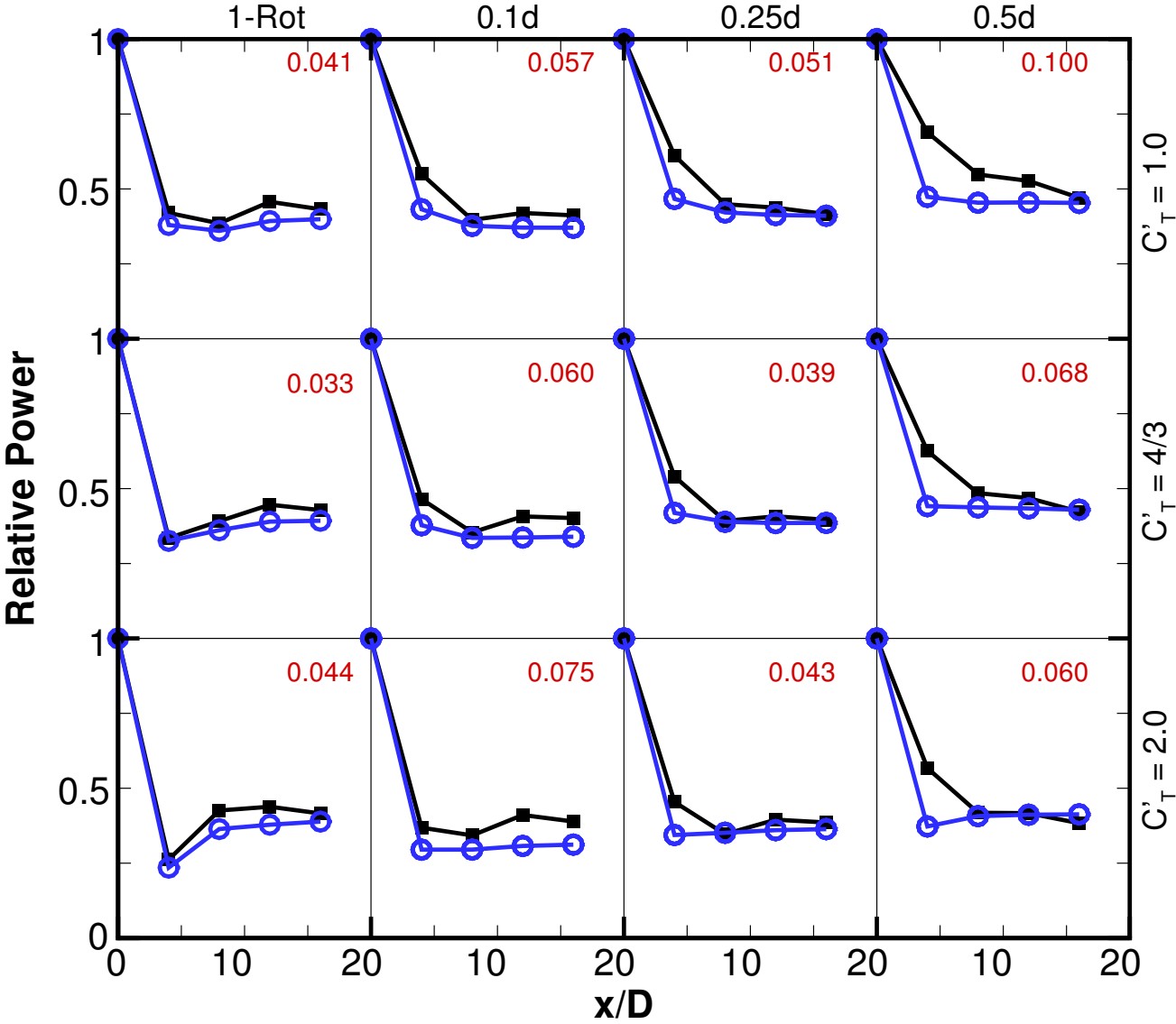

**Figure 15.** Relative power for 1-rotor and 4-rotor wind farms with fixed $S_X = 4D$, and varying tip spacing and thrust coefficient. Black squares are LES results. Blue circles are model predictions. Numbers in red are absolute errors in relative power averaged over turbines 2 through 5, $((1/4)\sum_{i=2}^{5}|(P_i/P_1)^{LES} - (P_i/P_1)^{model}|)$.

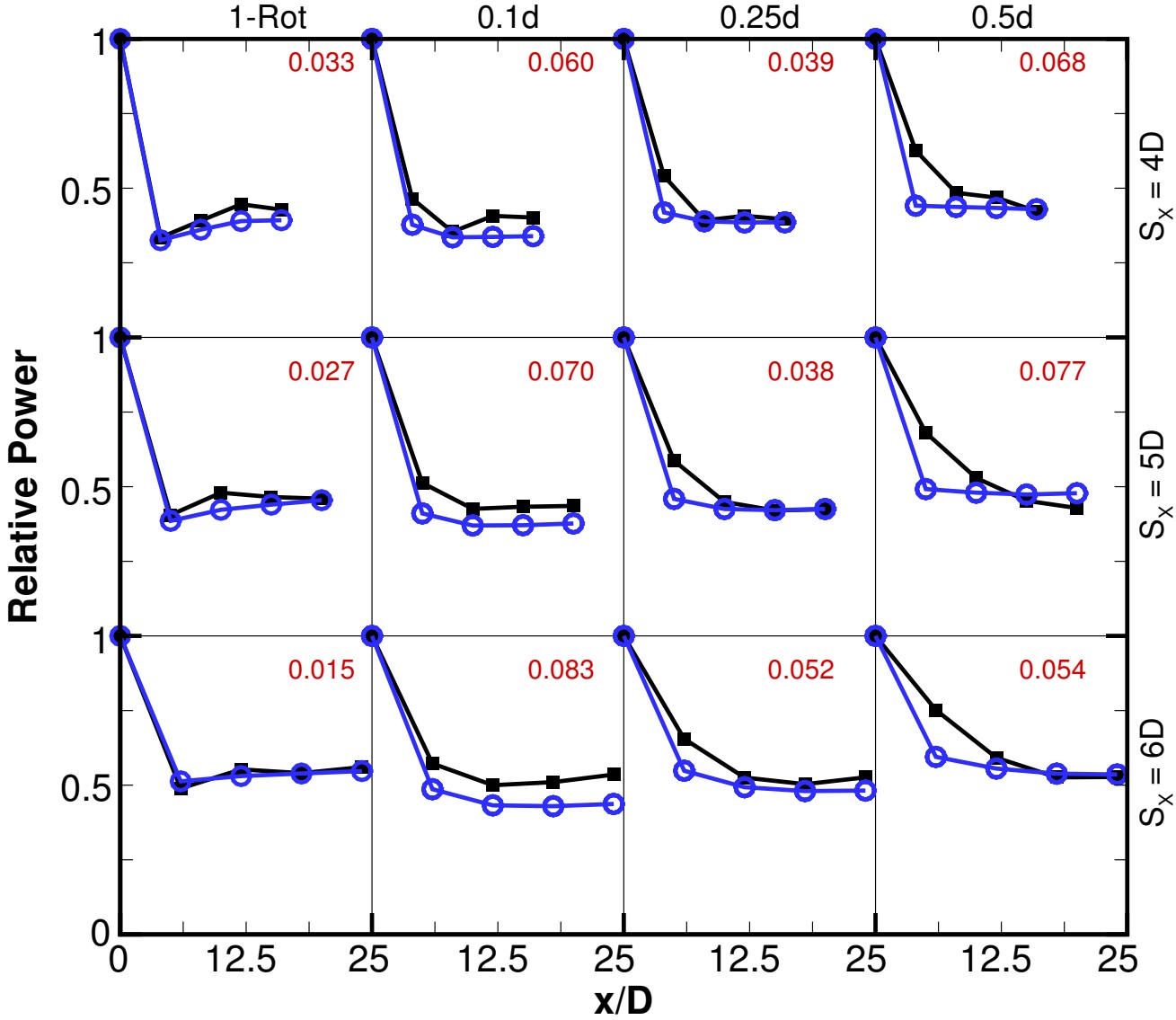

**Figure 16.** Relative power for 1-rotor and 4-rotor wind farms with fixed $C_T' = 4/3$, and varying tip spacing and axial spacing. Black squares are LES results. Blue circles are model predictions. Numbers in red are absolute errors in relative power averaged over turbines 2 through 5, $((1/4) \sum_{i=2}^{5} |(P_i/P_1)^{LES} - (P_i/P_1)^{model}|)$.

rotor is characterized by the smaller length scale, $d$. For realistic values of $s/d \sim 0.1 - 0.5$, the rotor wakes do not act entirely independent of each other, and the wakes do interact and merge with each other beyond a certain downstream distance. The reduced TKE levels suggest potential for reduced fatigue loads on the blades. These results for an isolated turbine are shown to be consistent for all thrust coefficient ($C_T'$) values evaluated.

In Sect. 4, a line of 5 turbines is evaluated to study the interaction between several multi-rotor wind turbines. For these wind farm simulations, the axial spacing ($S_X$) between different turbines is an important parameter, in addition to the tip spacing and the thrust coefficient. Consistent with the results of the isolated turbine LES, the velocity deficits are smaller in 4-rotor wind farms than in the corresponding 1-rotor wind farms until a certain distance into the wind farm. This distance increases with increasing $s/d$ and decreasing $S_X$. The turbulence intensity levels are significantly smaller for all downstream locations, which indicates potentially smaller fatigue loads for downstream turbines, for all combinations of $s/d$ and $S_X$. These results are, again, consistent for all $C_T'$ values evaluated using LES.

The effect of smaller velocity deficits is reflected in the relative powers, or equivalently, the wake losses experienced by wind farms. Wind farms comprised of multi-rotor turbines always show benefits over similar wind farms comprised of 1-rotor turbines. The benefits are due to smaller wake losses only for the first downstream turbine (i.e. the second turbine in the array) for a realistic tip spacing of $0.1$ times the diameter of the smaller rotor. The benefit increases with increasing tip spacing, and decreasing thrust coefficient. The benefit is largest for the smallest axial spacing studied here ($4D$), but does not decrease monotonically as the axial spacing is increased. The benefit is slightly larger for the largest axial spacing ($6D$) than for the intermediate spacing ($5D$). The effect of axial spacing on the benefit should be investigated in more detail in the future.

The analytical model predictions are sensitive to the tunable parameters. The results in Sect. 3.4 and 4.3 show that with appropriate choices, reasonably accurate predictions of the LES results can be obtained. The predictions are quite accurate beyond approximately $2D$ downstream of an isolated 1-rotor or 4-rotor turbine. In multi-turbine cases, the predictions are accurate for 1-rotor wind farms, and most 4-rotor wind farms. The model, however, fails to reproduce the trend of gradual decrease in relative power with turbine row, which is particularly pronounced for wind farms with larger $s/d$. The difficulties in accurately reproducing these trends are partly due to the fact that the Gaussian wake model is valid only beyond a certain distance downstream of a turbine, and is not valid immediately upstream and immediately downstream of a turbine. Thus, this study points to the need for better analytical modeling of the region very close (upstream as well as downstream) to the turbine.

The actuator drag-disk model provides a crude representation of the processes occurring very near the turbine disks. While this crude representation is sufficient for the purposes of capturing the interactions between the turbines and the atmospheric boundary layer, future studies should focus on using the actuator disk/line models with rotation of the blades included. Potential benefits associated with co-rotation and counter-rotation of the rotors in the multi-rotor configuration can be studied. Recent work by Andersen and Ramos-Garcia (2019) suggests that interaction between tip vortices of the individual rotors of the multi-rotor turbine aids in breakdown and recovery of the wake. These beneficial interactions might be missing from multi-rotor turbines with very large tip spacings, thus slowing down the rate of wake recovery. This issue can also be studied in the future. Fatigue loads on individual blades of isolated multi-rotor turbines as well as multi-rotor turbines downstream of other turbines

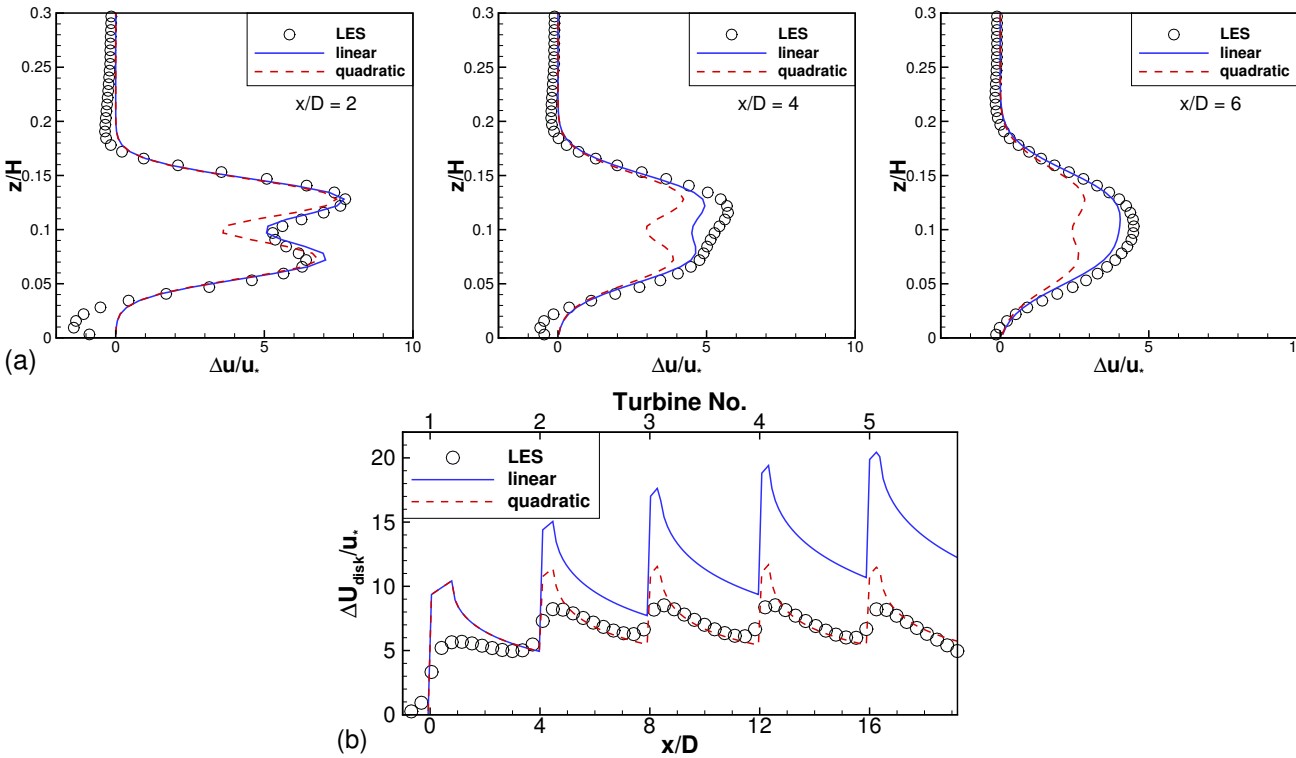

**Figure A1.** Evaluation of linear and quadratic wake merging methods for (a) isolated turbine with $(s/d, C_T') = (0.1, 4/3)$ and (b) wind farm with $(s/d, C_T', S_X) = (0.1, 4/3, 4D)$. Model parameter $\sigma_0/D = 0.28$, and $k_*$ values are the same as those for Figures 5 and 13(c) for panels (a) and (b) respectively.

should also be studied in the future. Finally, developing better analytical models for both, 1-rotor and multi-rotor, configurations continues to be a persistent challenge in wind energy research, and will be pursued in future work.

## Appendix A: Hybrid Linear-Quadratic Wake Superposition Methodology

A brief justification for following the hybrid linear-quadratic methodology of wake merging is provided in this appendix.

Figure A1(a) shows LES results and model predictions for the mean velocity deficit profiles for an isolated $s/d = 0.1$ turbine with $C_T' = 4/3$. Following the notation introduced in eq. (3), $N_{xt} = 1$ and $N_r(1) = 4$ for this case. The choices evaluated here are

$$(\Delta\bar{u}_{lin})_1 = \left[\sum_{j=1}^{N_r(1)} (\Delta\bar{u}_j(x,y,z))^p\right]^{1/p},$$

with $p = 1$ and 2 corresponding to linear and quadratic merging, respectively. It is clear that linear merging gives better agreement with LES results compared to quadratic merging. Thus, for wakes originating at the same $x$ location (i.e. 'adjacent' wakes), linear merging is preferred.

Figure A1(b) compares LES results and model predictions for the $s/d - 0.1$, $C_T' = 4/3$ and $S_X = 4D$ wind farm. Here, linear superposition of adjacent wakes is assumed, and superposition of these combined wakes originating at different $x$ locations is examined. The choices evaluated here are

$$\Delta \bar{u}_{tot} (x, y, z) = \left[ \sum_{i=1}^{N_{xt}} (\Delta \bar{u}_{lin})_i^p \right]^{1/p},$$

with, once again, $p = 1$ and 2 corresponding to linear and quadratic merging. For this case, $N_{xt} = 5$ and $N_r = 4$ for all $x_t$. Figure A1(b) shows that linear merging ($p = 1$) leads to a continuous increase of the velocity deficits, which is unphysical. Quadratic merging leads to velocity deficits that saturate a few turbines into the wind farm, and is in better qualitative and quantitative agreement with the LES results. Thus, quadratic merging is preferred for wakes originating at different $x$ locations.

Thus, a hybrid linear-quadratic merging strategy is seen to give best results. It should be noted that this is an empirical choice, and a physics-based/first-principles approach for wake superposition is a topic of active research.

### Appendix B: Potential Power of Multi-Rotor Wind Turbines

Finding an appropriate single-rotor turbine, which can be considered as a reference against which a multi-rotor turbine can be compared, is not straightforward. This is because the lower and upper pair of rotors in the 4-rotor configuration are subjected to different wind speeds and turbulence levels as compared to each other and to the single rotor in the 1-rotor configuration. In this work, we consider a single-rotor turbine with the same total frontal area, same thrust coefficient and same mean hub height as a multi-rotor turbine to be a reference. To test the appropriateness of this assumption, the potential power, computed as $P_{pot} = \left( \pi D^2 / 8 \right) C_P U_{0,disk}^3$, is shown in Table B1. Here, $U_{0,disk}$ is obtained by averaging the logarithmic inflow profile (shown in Fig. 2a) over the rotor disks. The potential power normalized by that of the 1-rotor turbine, $P_{pot} / P_{pot}^{1-Rot}$, is also shown in Table B1. A representative value of $C_P = 0.5625$ is used, but this precise number does not matter when we compare the normalized potential powers. The normalized potential powers are seen to be almost equal to 1 for all the tip spacings, and slightly reduce as the tip spacing increases. This indicates that the net effect of shear and the chosen dimensions of the turbines is such that the effect of the reduced wind speed seen by the lower two rotors dominates the effect of th larger wind speed seen by the upper two rotors. This effect is not very strong, being only $2.4\%$ for $s/d = 0.5$. For $s/d = 1$, the effect is larger, at $5.5\%$. The same conclusion is reached if we use the hub height velocities instead of the disk-averaged velocities in computing $P_{pot}$. For the present study, the chosen 1-rotor configuration may be considered to be appropriate as a reference, since its potential power varies by less than $2.4\%$ for the majority of the multi-rotor configurations.

**Table B1.** Potential power and potential power normalized by 1-rotor potential power for isolated turbines with varying tip spacings.

| s/d | 1-Rot | 0 | 0.05 | 0.1 | 0.2 | 0.25 | 0.5 | 1.0 |
|---|---|---|---|---|---|---|---|---|
| $P_{pot}$ | 11.21 | 11.17 | 11.15 | 11.13 | 11.09 | 11.07 | 10.95 | 10.59 |
| $P_{pot}/P_{pot}^{1-Rot}$ | 1.000 | 0.996 | 0.995 | 0.993 | 0.989 | 0.987 | 0.976 | 0.945 |

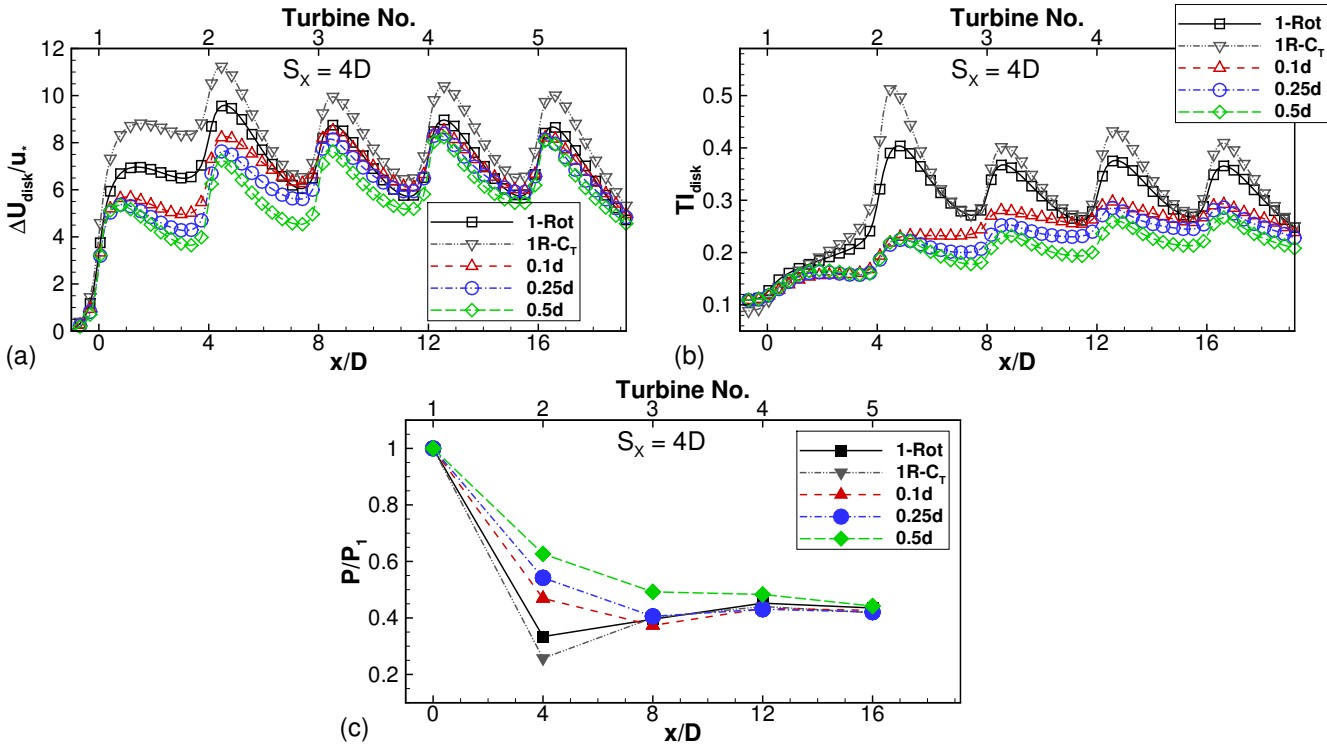

**Figure C1.** Adding results of '$C_T$-matched' run to Figures 9(a-c). Disk-averaged (a) velocity deficits, and (b) turbulence intensity, and (c) relative power for wind farms with axial spacing $S_X = 4D$. $C'_T = 1.61$ for simulation labeled $1R - C_T$ and $C'_T = 4/3$ for all other simulations.

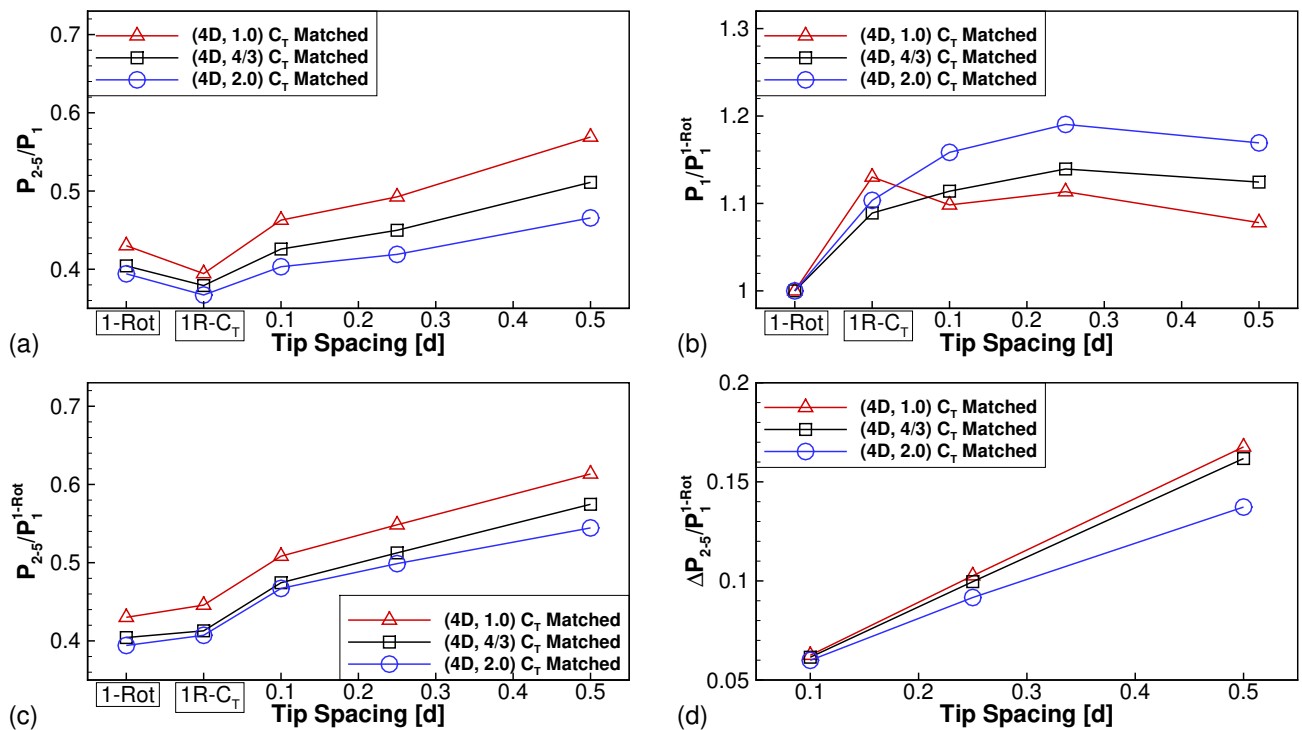

**Figure C2.** Adding results of '$C_T$-matched' runs to Fig. 11. Effect of tip spacing and thrust coefficient on (a) power of turbines 2 through 5 normalized by power of front turbine, (b) power of front turbine and (c) power of turbines 2 through 5 normalized by power of front turbine of corresponding 1-rotor wind farm. (d) Benefit of 4-rotor farms over corresponding $C_T$-matched 1-rotor wind farm. Labels indicate $(S_X, C_T')$ pairs. $C_T' = 1.14, 1.61$ and $2.47$ for the runs labeled $1R - C_T$, corresponding to $C_T' = 1, 4/3$ and $2$ respectively.

## Appendix C: $C_T$-Matched 1-Rotor Wind Farms

Single-rotor and multi-rotor turbines with the same rotor area, same mean hub height and same thrust coefficient have been considered to be equivalent in the main body of this paper. This equivalence was based on the 'local' thrust coefficient, $C_T'$. Assuming validity of the inviscid actuator-disk theory, imposing a local thrust coefficient implies imposing an induction factor, $a$, and a thrust coefficient, $C_T$. These quantities are related by

$$C_T' = \frac{C_T}{(1-a)^2}, \quad a^2 - a + \frac{C_T}{4} = 0. \tag{C1}$$

The classical actuator-disk theory, however, is not valid for the turbine disks subjected to the sheared, turbulent boundary layer inflow in this study. Consequently, given a value of $C_T'$, the implied values for $a$ and $C_T$ are different from those predicted by eq. (C1). Furthermore, since the single rotor in a 1-rotor turbine and the four individual rotors in a 4-rotor turbine are subjected to different values of shear and turbulence intensity, the implied values of $a$ and $C_T$ are different for the 1-rotor and 4-rotor

turbines. As seen in Fig. 11(b), the power of the front turbine in 1-rotor and 4-rotor wind farms is different although identical $C'_T$ values are used for all rotors.

In this appendix, three additional 1-rotor wind farm simulations are reported with $S_X = 4D$ and with $C'_T$ adjusted such that the resulting $C_T$ is closer to those of the corresponding 4-rotor turbines. Through a trial-and-error approach, $C'_T = 1.14$, 1.61 and 2.47 were found to yield $C_T$ values that are within $1.5\%$ of those of the 4-rotor wind farms with $C'_T = 1$, $4/3$ and $2$, respectively. These simulations are denoted as '$C_T$-matched' runs, and are labeled as $1R - C_T$ in Figures C1 and C2 here.

Figure C1 is a reproduction of Fig. 9(a-c) appended with the additional 1-rotor wind farm simulation with $C'_T = 1.61$. The disk-averaged velocity deficit and turbulence intensity profiles are larger than for the 1-rotor wind farm, particularly at $x/D = 4$ (turbine 2). The resulting power degradation (Fig. C1c) is more severe at turbine 2, and almost identical to the 1-rotor wind farm for further downstream turbines.

Figure C2 is a reproduction of Fig. 11 appended with results from all three '$C_T$-matched' runs. Focusing on the black line with squares in Fig. C2(b), it is seen that the power of the front turbine in the additional 1-rotor wind farm simulation (labeled '1R-$C_T$') is much closer to the powers of the front turbines in the three 4-rotor wind farms, than the front-turbine power in the 1-rotor simulation. In particular, the front-turbine power of the 4-rotor wind farm with $s/d = 0.25$ exceeds the front-turbine power of the '$C_T$-matched' wind farm by only $4.4\%$, while it exceeds the front-turbine power of the 1-rotor wind farm by almost $14\%$. Similarly, the front-turbine powers of the '1R-$C_T$' runs are much closer to those of the corresponding 4-rotor wind farms, than the front-turbine powers of the corresponding 1-rotor wind farm. Figures C2(a), (c) and (d) show the same qualitative behavior as Figures 11 (a), (c) and (d). In particular, the benefits of 4-rotor wind farms over the corresponding $C_T$-matched 1-rotor wind farms are seen in Fig. C2(d). This figure is derived from Fig. C2(c) by subtracting corresponding '1R-$C_T$' data point values from each of the 4-rotor data points. Although the numerical values are slightly different from Fig. 11(d), it is clear that the qualitative conclusions do not change, viz. the benefits of 4-rotor wind farms increase with increasing tip spacing and decreasing thrust coefficient.

In summary, this appendix ensures that the qualitative conclusions regarding the benefits of the 4-rotor wind farms remain unchanged, regardless of whether '1-Rot' ($C'_T$-matched) or '1R-$C_T$' ($C_T$-matched) 1-rotor wind farms are used for reference.

*Code and data availability.*  The LES code used for these simulations is available on GitHub at https://github.com/FPAL-Stanford-University/PadeOps. Data can be made available upon request from the corresponding author.

*Author contributions.*  All authors jointly designed the numerical experiments and interpreted the results. NSG and ASG wrote the code and performed the simulations. NSG prepared the manuscript with contributions from all authors.

*Competing interests.*  The authors declare that they have no conflict of interest.

*Acknowledgements.* Computational resources on TACC's Stampede2 cluster via NSF XSEDE Research Allocation TG-ATM170028 and on Stanford HPCC's Certainty cluster are gratefully acknowledged.

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
