# Peer review of "Effect of tip spacing, thrust coefficient and turbine spacing in multi-rotor wind turbines and farms"

_Wind Energy Science, 2019_

## Referee Comment (RC1) · Paul van der Laan (Referee) · 25 Jul 2019

**Review of *Effect of tip spacing, thrust coefficient and turbine spacing in multi-rotor wind turbines and farms* by **Niranjan S. Ghaisas, Aditya S. Ghate, and Sanjiva K. Lele.**

Reviewer: M. Paul van der Laan, DTU Wind Energy

The authors employ large-eddy simulations (LES) to quantify the difference in wake effects between a four-rotor wind turbine and an equivalent single-rotor wind turbine, for both an isolated wind turbine and a row of five wind turbines. The present work is a continuation of a conference article, where more parameters are investigated. An engineering wake model is calibrated with the LES data and its performance for a row of five four-rotor wind turbine is investigated. The article is well written and provides interesting results. I have written a list of main and minor comments below:

**Main comments**

1. Pages 1-2, Lines 57-64. You explain the difference choices compared to van der Laan et al. (2019), however, there are some misunderstandings:

    (a) You are right that in van der Laan et al. (2019) the thrust coefficient was different between the top and bottom rotors due to the shear and the rotor interaction, and the thrust force distribution was non-uniform for most of the published results by using body forces based on airfoil data. However, in the wake recovery analysis (which you focus on in this work) based on the Reynolds-averaged Navier-Stokes model, the same thrust force distribution and total thrust force was prescribed for the multi-rotor wind turbine and the equivalent single-rotor wind turbine, in order to make the comparison of the wake recovery more fair.

    (b) The inflow in van der Laan et al. (2019) is logarithmic, which corresponds to a neutral atmospheric surface layer (where the boundary height is infinite and the flow solution is independent of the Reynolds number). This means that the flow solution scales with the rotor diameter ($d$) and hub or multi-rotor reference height ($H_t$). In other words, the normalized wake deficit for using $d = 29.2$ m or $d = 50$ are the same, as long as $H_t$ is also up-scaled accordingly and the turbulence intensity, thrust coefficient, and tip clearances are the same. However, you can mention in your work, that you actually use a finite boundary layer height, instead of logarithmic profile with an infinite boundary layer height, as used in van der Laan et al. (2019).

2. You lack a recent reference to van der Laan and Abkar (2019), where the work of van der Laan et al. (2019) has been extended to multi-rotor wind farms. (The work was presented at the Wake Conference 2019 in May, but has been published online with some delay, in July, so you probably have missed this.) In this article, simulations suggest that multi-rotor wind farms produce 0.3-1.7% more annual energy production compared to equivalent single-rotor wind farms. The increase is mainly caused by the first downstream wind turbine in a wind farm and for row-aligned wind directions. This is in contradiction with your previous work Ghaisas et al. (2018) and it is related to the large rotor spacing of $1d$ that you had chosen, as shown in van der Laan and Abkar (2019). I think you should reference this work and highlight (in the introduction) that the difference between the single-rotor wind farm and multi-rotor wind farm in Ghaisas et al. (2018) was large due to the chosen tip clearance of $1d$. You could mention that the present work is used to investigate more realistic tip clearances. It is nice to see that you get similar trends in the power deficit (Figures 9 and 10), as published in van der Laan and Abkar (2019).

3. Page 4, Line 79: The relation $C_T = 16C_T'/(C_T' + 4)^2$ is only true for an axial induction factor $a_x$ of 1/3, since 1D Momentum Theory for a single-rotor in a uniform flow gives: $C_T = C_T'(1 - a_x)^2$ (which you also show in equation A1). In Table I: you investigate three different thrust coefficients (0.64, 0.75, 8/9). For 1D Momentum Theory for a single-rotor in a uniform flow we have that $C_T = 4a_x(1 - a_x)$ or $a_x = 1/2(1 - \sqrt{1 - C_T})$. Hence, for $C_T = 0.64, 0.75, 8/9$ we get $a_x = 0.2, 0.25, 1/3$, respectively. I would replace $C_T = 16C_T'/(C_T' + 4)^2$ with equation A1.

4. Lines 275-277: When you change the tip clearance, the amount of potential power changes due to the shear. Have you investigated how this affects your results? My guess is that the shear is the main reason why the power of the first wind turbine changes with different tip clearances (as you shown in Figure 11a).

5. Figure 6c: The single-rotor wind turbine seems to have a much higher disk averaged ambient turbulence intensity. Why is this the case? When I looked at the integrated added wake turbulence intensity, I found that the four-rotor wind turbine has a higher integrated added wake turbulence intensity in the near wake compared single-rotor wind turbine, while the opposite is found for the far wake, see Figure 18 in van der Laan et al. (2019).

6. Pages 18-22: Engineering wake model vs LES:

   (a) Please motivate the use of a spatial varying wake expansion parameter $k_*$.

   (b) Do I understand correctly, that you only fit $k_*$ from the single-rotor wind farm simulations and use this directly to predict the deficits in a multi-rotor wind farm (Figure 13)?

   (c) Figures 21 and 22: Please define the numbers colored in red in the caption, I guess it is the relative error. How did you compute it? $(P_{2-5}^{\text{model}} - P_{2-5}^{\text{LES}})/P_{2-5}^{\text{LES}}$?

   (d) How large are the differences between the wake model and LES if you had used a constant $k_*$? This is a relevant question because it is the standard usage of the chosen wake model.

7. Page 20, Discussion: Some wording needs to be changed here:

   (a) Line 337: Please remove the word *novel*, since the four-rotor wind turbine design is not new.

   (b) Lines 338-340: Please remove *for the first time*, since there are several authors (including yourself) have investigated the four-rotor wind turbine design.

8. Line 354: Please change *between different units* to *between different wind turbines*, or do you mean something else?

9. Line 286 and Lines 363-364: What do you mean by *The effect of the axial spacing on the benefit is ambiguous, since it is non-monotonic.*?

10. Appendix A: It is nice that you have added this appendix in order to make the comparison of a single-rotor and multi-rotor wind farm more fair, but you forgot to refer to it in the text.

**Minor comments**

1. Figures 5, 14 and 15 have very large labels compare to the rest of the figures. I would look nicer to keep the same label size.

2. You have normalized the velocity deficits by $u_*$, but this makes it hard to see how large the deficit actually is. It would be more interesting to normalize by the freestream velocity (which could be an integral over the disk area). When you normalize the turbulent kinetic energy by $u_*^2$, you could instead plot the turbulence intensity or added wake turbulence intensity, which is more common for wind turbine wake studies.

3. You both mention thrust coefficient and local thrust coefficient when you talk about $C_T'$. I would stick with local thrust coefficient everywhere to avoid confusion.

4. Figures 8, 9, 10, 12 and 13: I would write the wind turbine number (1, 2, 3, 4, 5) on the $x$-axis instead of $x/D$. This also corresponds better to the text, because you often talk about wind turbine numbers.

5. Figure 8: There are additional numbers plotted in Figure 8a.

6. You could refer to Niayifar and Porté-Agel (2015) when you talk about the relation between the local turbulence intensity and the wake recovery parameter $k_*$. Niayifar and Porté-Agel (2015) derived a relation between the freestream turbulence intensity and $k_*$ based on LES data.

**References**

Ghaisas, N. S., Ghate, A. S. ., and Lele, S. K.: Large-eddy simulation study of multi-rotor wind turbines, Journal of Physics: Conference Series, 1037, 1, https://doi.org/10.1088/1742-6596/1037/7/072021, 2018.

Niayifar, A. and Porté-Agel, F.: A new analytical model for wind farm power prediction, J. Phys.: Conf. Ser., 625, 1–10, https://doi.org/10.1088/1742-6596/625/1/012039, 2015.

van der Laan, M. P. and Abkar, M.: Improved energy production of multi-rotor wind farms, J. Phys.: Conf. Ser., 1256, 1–11, https://doi.org/10.1088/1742-6596/1256/1/012011, 2019.

van der Laan, M. P., J., A. S., Ramos García, N., Angelou, N., Pirrung, G. R., Ott, S., Sjöholm, M., Sørensen, K. H., Vianna Neto, J. X., Kelly, M., K., M. T., and Larsen, G. C.: Power curve and wake analyses of the Vestas multi-rotor demonstrator, Wind Energy Science, 4, 251–271, https://doi.org/10.5194/wes-4-251-2019, 2019.

---

## Referee Comment (RC2) · Søren Juhl Andersen (Referee) · 3 Aug 2019

Review comments for "Effect of tip spacing, thrust coefficient and turbine spacing in multi-rotor wind turbines and farms" by Ghaisas et al. in Wind Energy Science, 2019.

Overall: The article present numerous LES of both single rotor and multi-rotor consisting of 4 turbines. The different turbine configurations are compared, including the effects of tip spacing in the multi-rotor, thrust coefficient asn turbine spacing for farm scenarios. Additionally, the authors compare to the analytical model by Bastankhah and Porté-Agel. The article follows a number of other recent articles on multi-rotors and provides new results. The article is generally well-written and the results are interesting, so the article is recommended for publication with revisions according to the comments below.

General comments:

1. Resolution and degree of detail.

- The number of grid points are given in Table 1. However, it would be beneficial to report what these values correspond to the actual spatial resolution. Please correct me if wrong, but as far as I can tell, the main grid of 256x128x160 grid points has a width of pi/2*1000m, i.e. the lateral discretization is 1570.8m/128 = 12.3 m. Same resolution in the vertical. This means that there are only 4 points for a single actuator disc and only 2 for a small rotor in the multi-rotor. Is this correct? Tip spacing clearings corresponding to approximately 0.6m, 1.2m, 2.5m and 3m are investigated. How are the effect of tip spacing properly resolved when the mesh is so coarse?

- Please rephrase your sentence in the conclusion stating: "are studied in detail for the first time.". This is stretching it too far in my opinion for several reasons:

a) It could be argued that the degree of detail was larger in the article by van der Laan et al. (2019) due to higher resolution and using actuator lines rather than actuator disc as well as changing thrust due to a more realistic controller. Likewise, several of the conclusions found here corroborates the findings of other previous studies, but your present article still has merit. Additionally, the majority of the conclusions investigate integral quantities, e.g. power or disk-averaged velocity deficits.

b) A recent article by van der Laan and Abkar (2019) also investigates multi-rotors in wind farms and find similar conclusions. Please include as reference and discuss when results are similar or different. This is mainly that the benefit of multi-rotors seems to vanish further into the farm, as seen in Figure 9(c)+(f). The authors should comment on this more, because it also explains why the analytical model ends up giving reasonable results further into the farm as it approaches the same level as for single rotor wind

farms. Therefore, the conclusion by the authors "Wind farms comprised of multi-rotor turbines always show benefits over similar..." is perhaps also stretching the conclusions a bit as it does not show a benefit from the 4th turbine onwards.

2. Effect of CT. The authors discuss how a constant CT is used as opposed to the varying thrust level seen in van der Laan et al.(2019). Please comment on what is more realistic. Part of the discussion from the appendix on how to assess to CT could also be included in the main text.

3. Wake superposition. Wake superposition is not a trivial task and the focus of much research. The authors state in p. 5, line 124 that a new hybrid gives the best results. However, please elaborate on this, because it appears somewhat arbitrary. Best by what metric? It would be beneficial to include a comparison in an Appendix.

4. Reference/Comparison. Finding the appropriate reference for comparing a multi-rotor with a single rotor is not necessarily straightforward. Increasing the tip spacing a lot, has several implications for the presented results:

a) The upper multi-rotor will effectively see a higher wind speed than the single turbine and multi-rotor with smaller tip spacing. This will affect all the reported power increases, e.g. in Fig. 11.

b) As the tip spacing is increased, the wake merging is delayed and the authors state in p. 9, line 185-186: "...behave independently up to increasingly larger downstream distances". However, that means that it essentially becomes a comparison of a single wake behind a large rotor versus the wake behind a single small rotor. It can be seen in Figure 6(a) which also looks as if they would almost coincide if scaled properly by the corresponding rotor diameter and inflow velocity. Therefore, it seems that the conclusion by the authors is that is is beneficial to separate the rotors as much as possible, e.g. p. 15, line 285 "The benefit of 4-rotor wind farms increases with increasing tip spacing...". However, doing so would remove the potential benefial interaction of the tip vortices, which makes the wake break down faster, and hence recover faster. The

authors state that "...the 4-rotor turbine allows for greater entrainment". This is correct, but part of the increase might simply be an artifact of the reference no longer being appropriate. The question is if the entrainment from the center is more beneficial than the wake interaction? For details of the wake flow and how the wake interact to facilitate a faster breakdown, please see the published presentation with DOI by Andersen and Ramos-Garcia from WESC, 2019.

5. Analytical Model - The text in p. 19, line 310-313 does not seem to match Fig. 13: "Fig. 13(b) also show a similar sensitivity to the value of sigma"? It appears that sigma=0.28 gives better results for the velocity deficit, but worse for the power. Please explain this, because power should be proportional to $U^3$.

Technical Corrections:

- p. 1, line 20. Please define the "planform energy flux"

- p. 2, line 28-29: I doubt the cubic scaling laws were first realized in 2012. Please rephrase or find older reference.

- p. 2, line 33: "Overwhelmed" appears a odd choice of word. Please rephrase.

- p. 2, line 46: It is a little unclear which article "this paper" refers to, i.e. van der Laan et al. (2019) or Chasapogiannis et al. (2014). For the former, it is not entirely correct that the study by van der Laan et al. only considered isolated multi-rotors as it shows how the wind farm area can be significantly reduced due to faster wake recovery which inherently deals with multiple multi-rotors. Please rephrase accordingly.

- p. 3, line 71+74: What are the "standard" here? Or what would the non-standard be? Perhaps it would be beneficial to elaborate on the simulations framework.

- p. 3, line 98-99: Please specify what this correspond to in physical time.

- p. 4, line 80: It is unclear to me how you simply state that the SGS stresses can simply be neglected? Does that mean you're effectivley turning of your SGS model?

Please clarify.

- p. 6, line 131-33: Does equation 4 not give the deficit, rather than "mean velocity"? Please define u_tot

- p. 8, Fig. 3: Please be consistent in plotting. The linewidth in the symbols change from left to right, i.e. symbols are less clear.

- p. 11, line 195: "Grazing" appears a odd choice of word. Please rephrase.

- p. 15, line 158-261: Please rephrase these sentences. It does not appear as if turbine 3 in Fig. 11(f) produce "appreciable larger" power than for a single rotor.

- p. 15, line 268-270: The authors state "It is seen that P2-5 is larger for all 4-rotor wind farms...". This is not correct. If you look at Figure 10(c) there is actually a cross-over for the 3rd turbine, where the single rotor produces more. Be careful, when you do the aggregate statistics, because it gets lost. Please rephrase.

- p. 16, Fig. 9: The axes on Fig. 9(a)+(d) seems wrong? If the velocity deficit is normalized, should the axes not be between 0 and 1?

- p. 17, Fig. 10: Please improve the figure. It is very difficult(impossible) to tell the lines apart as the symbols are so large that they cover the full vs broken lines. Comment on the cross-over at the 3rd turbine.

- p. 18, Fig. 11(b): Only three lines are visible. Please explain/comment in the text.

- p. 19, line 328: Typo. Correct to "reproduced".

- p. 20, Fig. 13: The axes on Fig. 9(a)+(d) seems wrong? If the velocity deficit is normalized, should the axes not be between 0 and 1?

- p. 21-22, Fig. 14-15: Please include explanation of the red values and how they are computed. Figures should be self-contained.

Additional references:

1. van der Laan, M. P. and Abkar, M.: Improved energy production of multi-rotor wind farms, J. Phys.: Conf. Ser., 1256, 1–11, https://doi.org/10.1088/1742-6596/1256/1/012011, 2019.

2. Andersen and Ramos-Garcia: Dynamic Analysis of the Multi-rotor: Performance and Wake, WESC, 2019 https://doi.org/10.5281/zenodo.3357790

---

## Author Comment (AC1) · 19 Aug 2019

**Responses to Reviewer Comments on "Effect of tip spacing, thrust coefficient and turbine spacing in multi-rotor wind turbines and farms"**

Niranjan S. Ghaisas, Aditya S. Ghate, Sanjiva K. Lele

**1  Response to Reviewer 1**

*The authors employ large-eddy simulations (LES) to quantify the difference in wake effects between a four-rotor wind turbine and an equivalent single-rotor wind turbine, for both an isolated wind turbine and a row of five wind turbines. The present work is a continuation of a conference article, where more parameters are investigated. An engineering wake model is calibrated with the LES data and its performance for a row of five four-rotor wind turbine is investigated. The article is well written and provides interesting results. I have written a list of main and minor comments below:*

We thank Dr. M. Paul van der Laan for his careful reading and constructive comments on the manuscript. Responses to his comments are given below.

*Main comments:*

*1.  Pages 1-2, Lines 57-64. You explain the difference choices compared to van der Laan et al. (2019), however, there are some misunderstandings:*

*(a) You are right that in van der Laan et al. (2019) the thrust coefficient was different between the top and bottom rotors due to the shear and the rotor interaction, and the thrust force distribution was non-uniform for most of the published results by using body forces based on airfoil data. However, in the wake recovery analysis (which you focus on in this work) based on the Reynolds-averaged Navier-Stokes model, the same thrust force distribution and total thrust force was prescribed for the multi-rotor wind turbine and the equivalent single-rotor wind turbine, in order to make the comparison of the wake recovery more fair.*

**Response:** We thank the reviewer for the clarification. The revised manuscript has been modified accordingly (around line 123).

*(b) The inflow in van der Laan et al. (2019) is logarithmic, which corresponds*

*to a neutral atmospheric surface layer (where the boundary height is infinite and the flow solution is independent of the Reynolds number). This means that the flow solution scales with the rotor diameter (d) and hub or multi-rotor reference height (Ht). In other words, the normalized wake deficit for using d = 29:2 m or d = 50 are the same, as long as Ht is also up-scaled accordingly and the turbulence intensity, thrust coefficient, and tip clearances are the same. However, you can mention in your work, that you actually use a finite boundary layer height, instead of logarithmic profile with an infinite boundary layer height, as used in van der Laan et al. (2019).*

**Response:** The revised manuscript has been modified as pointed out by the reviewer (lines 63-66).

*2. You lack a recent reference to van der Laan and Abkar (2019), where the work of van der Laan et al. (2019) has been extended to multi-rotor wind farms. (The work was presented at the Wake Conference 2019 in May, but has been published online with some delay, in July, so you probably have missed this.) In this article, simulations suggest that multi-rotor wind farms produce 0.3-1.7% more annual energy production compared to equivalent single-rotor wind farms. The increase is mainly caused by the first downstream wind turbine in a wind farm and for row-aligned wind directions. This is in contradiction with your previous work Ghaisas et al. (2018) and it is related to the large rotor spacing of 1d that you had chosen, as shown in van der Laan and Abkar (2019). I think you should reference this work and highlight (in the introduction) that the difference between the single-rotor wind farm and multi-rotor wind farm in Ghaisas et al. (2018) was large due to the chosen tip clearance of 1d. You could mention that the present work is used to investigate more realistic tip clearances. It is nice to see that you get similar trends in the power deficit (Figures 9 and 10), as published in van der Laan and Abkar (2019).*

**Response:** A citation to this paper is included in the revised manuscript. This paper is discussed on lines 46-52 and 274-276.

*3. Page 4, Line 79: The relation $C_T = 16C_T'/(C_T' + 4)^2$ is only true for an axial induction factor $a_x$ of 1/3, since 1D Momentum Theory for a single-rotor in a uniform flow gives: $C_T =$ (which you also show in equation A1). In Table I: you investigate three different thrust coefficients (0.64, 0.75, 8/9). For 1D Momentum Theory for a single rotor in a uniform flow we have that $C_T = 4a_x(1-a_x)$ or $a_x = 1/2\left(1 - \sqrt{1 - C_T}\right)$. Hence, for $C_T = 0 : 64; 0 : 75; 8/9$ we get $a_x = 0 : 2; 0 : 25; 1/3$, respectively. I would replace $C_T = 16C_T'/(C_T'+4)^2$ with equation A1.*

**Response:** We thank the reviewer for pointing out this oversight. It has been corrected in the revised manuscript.

*4. Lines 275-277: When you change the tip clearance, the amount of poten-*

Table 1: Potential power and potential power normalized by 1-Rotor potential power for isolated turbines with varying tip spacings.

| s/d | 1-Rot | 0 | 0.05 | 0.1 | 0.2 | 0.25 | 0.5 | 1.0 |
|---|---|---|---|---|---|---|---|---|
| $P_{pot}$ | 11.21 | 11.17 | 11.15 | 11.13 | 11.09 | 11.07 | 10.95 | 10.59 |
| $P_{pot}/P_{pot}^{1-Rot}$ | 1.000 | 0.996 | 0.995 | 0.993 | 0.989 | 0.987 | 0.976 | 0.945 |

*tial power changes due to the shear. Have you investigated how this affects your results? My guess is that the shear is the main reason why the power of the first wind turbine changes with different tip clearances (as you shown in Figure 11a).*

**Response:** The reviewer is correct in saying that the potential power is different for 4-rotor turbines with different tip clearances. Due to the sheared inflow velocity profile, the upper rotors of the 4-rotor turbine are subjected to a higher wind speed as compared to the wind speed seen by the 1-rotor turbine. Similarly, the lower rotors of the 4-rotor turbine see a lower wind speed. Thus, the potential power is expected to be dependent on the tip spacing, since the undisturbed velocity seen by the rotors is dependent on the tip spacing. However, we find that this does not explain the differences between the powers of the first turbine shown in Fig. 11b, as explained in more detail below.

The potential power is computed as $P_{pot} = (\pi D^2/8)C_P U_{0,disk}^3$, where $U_{0,disk}$ is obtained by averaging the logarithmic profile (shown in Fig. 2a) over the rotor disks. $P_{pot}$ and $P_{pot}/P_{pot}^{1-Rot}$, or the potential power normalized by that of the 1-rotor isolated turbine, are shown in Table 1. A representative value of $C_P = 0.5625$ is used, but this precise number does not matter when we compare the normalized potential powers.

The normalized potential powers can be seen to be almost equal to 1 for all the tip spacings. In fact, the normalized potential powers reduce slightly as the tip spacing increases. The net effect of shear and the chosen dimensions of the turbines is such that the reduction in power of the lower two rotors dominates the increase in power of the upper two rotors. This effect is not very strong, as seen in the Table 1, being only 2.4% for $s/d = 0.5$. The same conclusion is reached if we use the hub height velocities instead of the disk-averaged velocities in computing $P_{pot}$. Thus, the differences between the powers of the isolated 4-rotor turbines cannot be attributed to differences in the potential powers. These exact same observations hold for the first turbines in the multi-rotor wind farm cases (powers shown in Fig. 12b).

We hypothesize that the differences for different tip spacings is due to discrepancies between the simulations and the predictions of the inviscid actuator disk (AD) theory. The predictions of the AD theory and the current LES can be either due to inadequate resolution, or due to the effect of turbulent mixing (we refer to Nishino and Wilden (2012) for a discussion on the effect of mixing on thrust and power coefficients of a turbine), which is not accounted for in the AD theory. To rule out the effect of grid resolution on the results, the implied nom-

Table 2: Implied nominal thrust coefficient for different grid resolutions and two different cases.

| Grid Size | 1-Rotor | $s/d = 0.05$ |
|-----------|---------|--------------|
| GR1 | 0.890 | 0.971 |
| GR2 | 0.798 | 0.859 |
| GR3 | 0.799 | 0.853 |

inal thrust coefficient ($C_T$) is shown in Table 2 for the isolated 1-rotor case and the isolated 4-rotor case with $s/d = 0.05$. These values are calculated from the LES using the definition $C_T = C'_T U^2_{disk}/U^2_{0,disk}$.It is seen that these values are quite well converged. Changing the grid resolution from GR2 to GR3 changes $C_T$ by 0.1% for the 1-rotor case and 0.6% for the 4-rotor case. We conclude that the discrepancies with the inviscid actuator disk theory are due to the effects of turbulent mixing, which are different for different tip clearances. In any case, as also appreciated by the reviewer in a separate comment, the discussion in the appendix further clarifies that these differences between the first turbine powers do not affect the qualitative conclusions drawn. To keep the paper focused, we do not include the above discussion in detail in the main manuscript. We have added an appendix quantifying the change in potential power with changing tip spacings (Appendix B) and added a sentence in Sec. 4.2 (lines 309-311) stating that these discrepancies cannot be explained by the differences in power potential. The response to the reviewer in this present document is available online for readers interested in more details.

*5. Figure 6c: The single-rotor wind turbine seems to have a much higher disk averaged ambient turbulence intensity. Why is this the case?*

**Response:** We appreciate the reviewer's eye for detail. The larger ambient turbulence intensity seen in Fig. 6c actually corresponds to the 4-Rotor $s/d = 1$ case. We apologize for the confusion: the line styles chosen for the 1-Rotor and 4-rotor $s/d = 1$ cases were very similar to each other. In the updated manuscript, a different line style has been used for the $s/d = 1$ case.

It should be noted that the integrals are carried out over different regions corresponding to the areas spanned by the rotors in the different cases. Thus, the ambient $TI_{disk}$ values are slightly different for all cases. The differences are apparent on the scale of the figure only for the $s/d = 1$ case, which is why it appears to be an outlier.

*When I looked at the integrated added wake turbulence intensity, I found that the four-rotor wind turbine has a higher integrated added wake turbulence intensity in the near wake compared single-rotor wind turbine, while the opposite is found for the far wake, see Figure 18 in van der Laan et al. (2019).*

[Figure]

Figure 1: Added turbulence intensity $\Delta I_{disk}$ for isolated 1-rotor and 4-rotor turbines with varying tip spacings denoted in the label, for $C'_T = 4/3$.

**Response:** This is an interesting observation, that is not reproduced in our LES. It should be noted that the quantities plotted in our Fig. 6c and in Figs. 18(b,d,f) in van der Laan et al. (2019) are different. For a fair comparison between the two sets of results, the disk-averaged added turbulence intensity, defined as

$$\Delta I_{disk} = \frac{\sqrt{2TKE_{disk}/3}}{U_{disk}} - I_{0,disk}, \tag{1}$$

where

$$I_{0,disk} = \frac{\sqrt{2TKE_{0,disk}/3}}{U_{0,disk}}, \tag{2}$$

is plotted in Figure 1. The subscript 'disk' denotes disk-average and the subscript '0' denotes upstream region, or the region between $x/D = -4$ to $x/D = -1$ in our simulations, where the quantities are almost constant. This measure of added turbulence intensity is compared to Figs. 18(b,d,f) of van der Laan et al. (2019).

It is seen that $\Delta I_{disk}$ for all the 4-rotor cases is always smaller than for the 1-rotor case. The qualitative differences between the near-wake and far-wake regions are not observed in our LES results. A closer examination of the results in van der Laan et al. (2019) suggests that $\Delta I_{disk}$ is larger for the 4-rotor case in the near-wake region only for $I_{ref} = 5\%$ and 10%, but not for $I_{ref} = 20\%$. Thus, our results may be considered to be qualitatively similar to the results of van der Laan et al. (2019) for $I_{ref} = 20\%$. It is difficult to gauge the accuracy of one LES with another LES as a benchmark, so we refrain from commenting on which LES result is more accurate. This information is included in Fig. 6(d) of the revised manuscript and in Section 3.2 (lines 218-224).

*6. Pages 18-22: Engineering wake model vs LES:*

*(a) Please motivate the use of a spatial varying wake expansion parameter $k_*$.*

**Response:** Using a spatially constant $k_*$ always leads to a gradual drop in the relative power prediction. The recovery in relative power, which is typically observed after the second turbine in conventional 1-rotor wind farms and after the third turbine in the 4-rotor wind farms with $s/d = 0.1$ and $0.25$, is not seen in the model predictions using a constant $k_*$. This is the reason for preferring a framework where the value of $k_*$ is tied to the local turbulence intensity. While this framework relies on an empirical relation between $k_*$ and the local turbulence intensity, it yields better qualitative and quantitative results, and is preferred here. It is possible for individual 4-rotor wind farm cases (particularly those with $s/d = 0.5$) to be better predicted using a constant $k_*$ value in combination with some value for $\sigma_0/D$. However, we could not arrive at a simple consistent rule that yielded good predictions for all cases, and conclude that the spatially varying $k_*$ framework works better than a spatially-constant $k_*$. We have added Figure 14 to the revised manuscript and accompanying discussion to Sec. 4.3 (lines 358-371).

*(b) Do I understand correctly, that you only fit $k_*$ from the single-rotor wind farm simulations and use this directly to predict the deficits in a multi-rotor wind farm (Figure 13)?*

**Response:** Yes, the $k_*$ vs $I_x$ correlation is derived from only the 1-rotor LES results. This is done for the present to avoid complications arising from lateral merging of the wakes of the four rotors of one 4-rotor turbine and axial merging of wakes of different 4-rotor turbines.

*(c) Figures 21 and 22: Please define the numbers colored in red in the caption, I guess it is the relative error. How did you compute it?*

**Response:** The absolute error between the LES result and the model prediction of the relative power is computed at each of the four downstream turbines (turbines 2 through 5), and then averaged. Thus, the numbers in red in Figures 15 and 16 are $(1/4)\sum_{i=2}^{5} |(P_i/P_1)^{LES} - (P_i/P_1)^{model}|$. We do not normalize the absolute errors any further (say with the average relative power given by the LES) because it would lead to misleadingly small numbers for cases where the average relative power is large (for example, the $S_X = 4D$, $C_T' = 1$ case). The formula for the error is included in the revised manuscript on line 375 and the captions to Figures 15 and 16.

*(d) How large are the differences between the wake model and LES if you had used a constant $k_*$? This is a relevant question because it is the standard usage of the chosen wake model.*

**Response:** Please see response to point 6(a) above and the revised manuscript (Figure 14; Sec. 4.3, lines 358-371).

*7. Page 20, Discussion: Some wording needs to be changed here:*

*(a) Line 337: Please remove the word novel, since the four-rotor wind turbine design is not new.*

**Response:** The word has been removed.

*(b) Lines 338-340: Please remove for the first time, since there are several authors (including yourself) have investigated the four-rotor wind turbine design.*

**Response:** The phrase has been removed.

*8. Line 354: Please change between different units to between different wind turbines, or do you mean something else?*

**Response:** We did mean wind 'turbines'. The word has been replaced as suggested.

*9. Line 286 and Lines 363-364: What do you mean by The effect of the axial spacing on the benefit is ambiguous, since it is non-monotonic.?*

**Response:** As seen in Figure 11(d), the wake-related benefits of multi-rotor wind farms are largest for $S_X = 4D$ and smallest for $S_X = 5D$ (comparing cases with $C'_T = 4/3$ only). Thus, the benefits do not monotonically increase or decrease with $S_X$. This is clarified better in the revised manuscript (lines 416-418).

*10. Appendix A: It is nice that you have added this appendix in order to make the comparison of a single-rotor and multi-rotor wind farm more fair, but you forgot to refer to it in the text.*

**Response:** A reference to the appendix has been added in Sec. 4.2 of the revised manuscript (lines 322-323).

*Minor comments*

*1. Figures 5, 14 and 15 have very large labels compare to the rest of the figures. I would look nicer to keep the same label size.*

**Response:** The label sizes on these figures have been reduced in the revised manuscript.

*2. You have normalized the velocity deficits by $u_*$, but this makes it hard to see how large the deficit actually is. It would be more interesting to normalize by the freestream velocity (which could be an integral over the disk area). When*

*you normalize the turbulent kinetic energy by $u_*^2$, you could instead plot the turbulence intensity or added wake turbulence intensity, which is more common for wind turbine wake studies.*

**Response:** We agree with the reviewer that normalizing by the (disk-averaged) freestream velocity would be an interesting alternative way of plotting the results. The current way of normalizing by the friction velocity of the ABL is also valuable because it allows for comparison between the different cases presented in any one figure. The reader can compare with Figure 2 to get a sense for the magnitudes of the deficits with respect to the freestream velocity. The turbulence intensity is shown in Figures 6 and 7 for isolated turbine cases and Figures 9, 10 and A1 for multiple turbine simulations.

*3. You both mention thrust coefficient and local thrust coefficient when you talk about $C_T'$. I would stick with local thrust coefficient everywhere to avoid confusion.*

**Response:** We feel the need to introduce both thrust coefficients because the local thrust coefficient ($C_T'$) is used in the LES while the nominal thrust coefficient ($C_T$) is used in the analytical model. It would be awkward to denote $C_T$ in the model expressions without having introduced it earlier.

*4. Figures 8, 9, 10, 12 and 13: I would write the wind turbine number (1, 2, 3, 4, 5) on the x-axis instead of $x/D$. This also corresponds better to the text, because you often talk about wind turbine numbers.*

**Response:** Turbine numbers are added to Figures 8, 9, 10, 12, 13, 14 and A1 in the revised manuscript.

*5. Figure 8: There are additional numbers plotted in Figure 8a.*

**Response:** These numbers are included in Figure 8a (as well as in Figure 4a) deliberately, so as to draw attention to specific contour levels, as explained in the figure caption.

*6. You could refer to Niayifar and Porté-Agel (2015) when you talk about the relation between the local turbulence intensity and the wake recovery parameter $k_*$. Niayifar and Porté-Agel (2015) derived a relation between the freestream turbulence intensity and $k_*$ based on LES data.*

**Response:** We have referred to this paper in the revised manuscript (Sec. 4.3, lines 366-371).

*References*

*Ghaisas, N. S., Ghate, A. S ., and Lele, S. K.: Large-eddy simulation study*

of multi-rotor wind turbines, Journal of Physics: Conference Series, 1037, 1, https://doi.org/10.1088/1742-6596/1037/7/072021, 2018.

Niayifar, A. and Porté-Agel, F.: A new analytical model for wind farm power prediction, J. Phys.: Conf. Ser., 625, 1–10, https://doi.org/10.1088/1742-6596/625/1/012039, 2015.

van der Laan, M. P. and Abkar, M.: Improved energy production of multi-rotor wind farms, J. Phys.: Conf. Ser., 1256, 1–11, https://doi.org/10.1088/1742-6596/1256/1/012011, 2019.

van der Laan, M. P., J., A. S., Ramos García, N., Angelou, N., Pirrung, G. R., Ott, S., Sjöholm, M., Sørensen, K. H., Vianna Neto, J. X., Kelly, M., K., M. T., and Larsen, G. C.: Power curve and wake analyses of the Vestas multi-rotor demonstrator, Wind Energy Science, 4, 251–271, https://doi.org/10.5194/wes-4-251-2019, 2019.

**2    Response to Reviewer 2**

*Overall: The article present numerous LES of both single rotor and multi-rotor consisting of 4 turbines. The different turbine configurations are compared, including the effects of tip spacing in the multi-rotor, thrust coefficient asn turbine spacing for farm scenarios. Additionally, the authors compare to the analytical model by Bastankhah and Porté-Agel. The article follows a number of other recent articles on multi-rotors and provides new results. The article is generally well-written and the results are interesting, so the article is recommended for publication with revisions according to the comments below.*

**Response:** We thank the reviewer, Dr. S. J. Andersen, for his careful assessment of our manuscript and for the detailed and constructive comments.

*General comments:*

*1. Resolution and degree of detail.*

*- The number of grid points are given in Table 1. However, it would be beneficial to report what these values correspond to the actual spatial resolution. Please correct me if wrong, but as far as I can tell, the main grid of 256x128x160 grid points has a width of pi/2*1000m, i.e. the lateral discretization is 1570.8m/128 = 12.3 m. Same resolution in the vertical. This means that there are only 4 points for a single actuator disc and only 2 for a small rotor in the multi-rotor. Is this correct? Tip spacing clearings corresponding to approximately 0.6m, 1.2m, 2.5m and 3m are investigated. How are the effect of tip spacing properly resolved when the mesh is so coarse?*

**Response:** Assuming that the boundary layer height $H$ is 1000 m, the grid resolution for the main $256 \times 128 \times 160$ grid (G2) is $12.3 \times 12.3 \times 6.25$ m. The single-rotor turbine has a diameter of $D = 0.1H = 100$ m. Thus, we have approximately 8 grid points across the disk in the spanwise direction and 16 points across the disk in the vertical direction. For each of the rotors in the multi-rotor turbines, the diameter is $d = 0.05H = 50$ m, which leads to 4 and 8 grid points in the spanwise and vertical directions respectively. It should be noted that for the combined 4-rotor system, the number of grid points is again similar to that for the larger single-rotor configuration ($8 \times 16$). The dimensional values for the tip spacings are $(2.5, 5, 10, 12.5, 25, 50)$ m corresponding to $s/d = (0.05, 0.1, 0.2, 0.25, 0.5, 1)$, respectively.

Using upwards of 8 grid points across the disk is an established rule-of-thumb following the study by Wu and Porté-Agel (2011). Other studies have used smaller number of grid points across the disk, particularly in the spanwise direction (e.g. (Stevens et al., 2014) used only 4 points across the disk in the spanwise direction in their main grid A3). Thus, the resolution used here with grid G2 is consistent with previous studies. In addition, the grid independence study in Section 3.1 of our manuscript quantifies the change in results between

grids G2 and G3. The change in mean velocities is marginal, while the change in added turbulent kinetic energy (TKE) is about 9 % near the top-tip height, and about 2% when averaged over the disk regions. We believe this level of convergence is sufficient to derive confidence in our results.

While the tip spacings for several $s/d$ values are smaller than the grid spacing in the spanwise direction, the effect of tip spacing is captured because the actuator disk model appropriately adjusts the distribution of the thrust force across the discretization points. The details between the tips are obviously missed with this coarse resolution, but the wake effects are appropriately captured. The grid independence study referred to above was carried out for the smallest non-zero spacing, $s/d = 0.05$, as well as for the largest $s/d = 1$, and showed similar level of convergence. Details of the grid dependence for the $s/d = 1$ case are not shown in this manuscript for brevity, but may be found in Ghaisas et al. (2018).

*- Please rephrase your sentence in the conclusion stating: "are studied in detail for the first time.". This is stretching it too far in my opinion for several reasons:*

*a) It could be argued that the degree of detail was larger in the article by van der Laan et al. (2019) due to higher resolution and using actuator lines rather than actuator disc as well as changing thrust due to a more realistic controller. Likewise, several of the conclusions found here corroborates the findings of other previous studies, but your present article still has merit. Additionally, the majority of the conclusions investigate integral quantities, e.g. power or disk-averaged velocity deficits.*

**Response:** We agree with the reviewer that some of the findings here corroborate those in van der Laan et al. (2019). The phrase pointed out has been removed.

*b) A recent article by van der Laan and Abkar (2019) also investigates multi-rotors in wind farms and find similar conclusions. Please include as reference and discuss when results are similar or different. This is mainly that the benefit of multi-rotors seems to vanish further into the farm, as seen in Figure 9(c)+(f). The authors should comment on this more, because it also explains why the analytical model ends up giving reasonable results further into the farm as it approaches the same level as for single rotor wind farms. Therefore, the conclusion by the authors "Wind farms comprised of multi-rotor turbines always show benefits over similar..." is perhaps also stretching the conclusions a bit as it does not show a benefit from the 4th turbine onwards.*

**Response:** We thank the reviewer for pointing out this new article. In the revised manuscript, we refer to this article in the introduction as well as in the results sections (lines 46-53 and 272-275). The statement that multi-rotor wind farms are always beneficial is correct in the sense that the relative power averaged over all downstream rows is larger for multi-rotor farms as compared

to for single-rotor farms because the relative power of the first downstream row is always larger in the multi-rotor farm compared to in the single-rotor farm. However, we agree with the reviewer that this needs to be qualified, so a statement to the effect that the benefit is only due to the first downstream turbine in realistic tip spacing cases is included in the revised manuscript (lines 414-415).

*2. Effect of CT. The authors discuss how a constant CT is used as opposed to the varying thrust level seen in van der Laan et al.(2019). Please comment on what is more realistic. Part of the discussion from the appendix on how to assess to CT could also be included in the main text.*

**Response:** A constant $C_T$ vs varying thrust level occurs in two respects. First, the thrust coefficient is fixed in time in our simulations, while in van der Laan et al. (2019), pitch and torque controllers are adopted in the simulations, which effctively lead to dynamically varying force coefficients. This information is included in Sec. 1 (lines 67-70) in the revised manuscript. Second, field measurements and related simulations in van der Laan et al. (2019) show that the thrust coefficients are different between the top and bottom pairs of rotors in the multi-rotor configuration, while identical thrust coefficients are used for all rotors in our simulations. This information is included in the revised manuscript in Sec. 2.2 (lines 120-124). Following comments from the other reviewer of this manuscript (Dr. M. Paul van der Laan), we realize that the simulations in van der Laan et al. (2019) also impose identical forcing to all rotors of the multi-rotor turbine for the purposes of comparing the wake recovery features.

We could not find a way to incorporate the material in the appendix into the main manuscript without taking the focus away. Hence, we choose to retain the material in the appendix. A reference to the appendix was missing from the main text of the original manuscript, which has now been included in Sec. 4.2 (lines 309-311).

*3. Wake superposition. Wake superposition is not a trivial task and the focus of much research. The authors state in p. 5, line 124 that a new hybrid gives the best results. However, please elaborate on this, because it appears somewhat arbitrary. Best by what metric? It would be beneficial to include a comparison in an Appendix.*

**Response:** We agree with the reviewer that wake superposition is an important topic of research currently. We have added an appendix to the manuscript to elaborate on why we use the hybrid method in this manuscript. The appendix shows that linear superposition of adjacent wakes is better than quadratic superposition, and that quadratic superposition of downstream wakes is better than linear superposition.

*4. Reference/Comparison. Finding the appropriate reference for comparing a multirotor with a single rotor is not necessarily straightforward. Increasing the tip spacing a lot, has several implications for the presented results:*

*a) The upper multi-rotor will effectively see a higher wind speed than the single turbine and multi-rotor with smaller tip spacing. This will affect all the reported power increases, e.g. in Fig. 11.*

*b) As the tip spacing is increased, the wake merging is delayed and the authors state in p. 9, line 185-186: "...behave independently up to increasingly larger downstream distances". However, that means that it essentially becomes a comparison of a single wake behind a large rotor versus the wake behind a single small rotor. It can be seen in Figure 6(a) which also looks as if they would almost coincide if scaled properly by the corresponding rotor diameter and inflow velocity. Therefore, it seems that the conclusion by the authors is that is is beneficial to separate the rotors as much as possible, e.g. p. 15, line 285 "The benefit of 4-rotor wind farms increases with increasing tip spacing...". However, doing so would remove the potential beneficial interaction of the tip vortices, which makes the wake break down faster, and hence recover faster. The authors state that "...the 4-rotor turbine allows for greater entrainment". This is correct, but part of the increase might simply be an artifact of the reference no longer being appropriate. The question is if the entrainment from the center is more beneficial than the wake interaction? For details of the wake flow and how the wake interact to facilitate a faster breakdown, please see the published presentation with DOI by Andersen and Ramos-Garcia from WESC, 2019.*

**Response:** We thank the reviewer for pointing out the interesting study on the interaction between tip-vortices of adjacent rotors in the multi-rotor configuration leading to a faster breakdown and recovery of the wake. We agree with all the points mentioned: (a) too large of a tip-spacing can lead to diminished benefits because it would reduce the interaction between the tip-vortices; (b) complications involved in designating a single-rotor configuration as being equivalent to a multi-rotor configuration.

With regards to point (a), we have mentioned this in Section 5 (lines 430-433). With regards to point (b), we have calculated the potential power of the multi-rotor turbine for different tip spacings. As discussed in a new appendix (Appendix B) in the revised manuscript, the potential power of the multi-rotor configuration differs from the potential power of the 1-rotor configuration by fairly small numbers (less than 2.4% for $s/d$ up to 0.5). The difference is 5.5% for $s/d = 1$, which is admittedly large. For the present study, the chosen 1-rotor configuration may be considered to be appropriate as a reference, since its potential power varies by less than 2.4% (a small, but admittedly arbitrary number) for the majority of the multi-rotor configurations. This information is included in Appendix B in the revised manuscript.

*5. Analytical Model - The text in p. 19, line 310-313 does not seem to match Fig. 13: "Fig. 13(b) also show a similar sensitivity to the value of sigma"? It appears that sigma=0.28 gives better results for the velocity deficit, but worse for the power. Please explain this, because power should be proportional to $U^3$.*

**Response:** The point we wanted to make here was that $\sigma_0/D = 0.28$ gives better results for the velocity deficit only in the region approximately $1D - 3D$ downstream of each turbine, but not close to the turbine. $\sigma_0/D = 0.32$ gives better prediction closer to the turbine. Thus, better modeling of the region at and close to the turbine is important for predicting power. We have reworded the paragraph to hopefully make these points more clear.

*Technical Corrections:*

**Response:**

*- p. 1, line 20. Please define the "planform energy flux"*

**Response:** We have replaced 'planform energy flux' with the more appropriate 'power density' and defined it in the first paragraph.

*- p. 2, line 28-29: I doubt the cubic scaling laws were first realized in 2012. Please rephrase or find older reference.*

**Response:** We tried to find an older reference (in papers and textbooks) without success. The sentence has been slightly rephrased.

*- p. 2, line 33: "Overwhelmed" appears a odd choice of word. Please rephrase.*

**Response:** The sentence has been rephrased.

*- p. 2, line 46: It is a little unclear which article "this paper" refers to, i.e. van der Laan et al. (2019) or Chasapogiannis et al. (2014). For the former, it is not entirely correct that the study by van der Laan et al. only considered isolated multi-rotors as it shows how the wind farm area can be significantly reduced due to faster wake recovery which inherently deals with multiple multi-rotors. Please rephrase accordingly.*

**Response:** We meant to refer to van der Laan et al. (2019). This sentence has been rephrased.

*- p. 3, line 71+74: What are the "standard" here? Or what would the non-standard be? Perhaps it would be beneficial to elaborate on the simulations framework.*

**Response:** We have included a reference to our previous work (Ghate and Lele, 2017) where equations and numerical details have been mentioned.

*- p. 3, line 98-99: Please specify what this correspond to in physical time.*

**Response:** The numbers are added to the revised manuscript.

*- p. 4, line 80: It is unclear to me how you simply state that the SGS stresses can simply be neglected? Does that mean you're effectivley turning of your SGS model? Please clarify.*

**Response:** We apologize for this confusion. This was a typo. 'Subgrid' has been corrected to 'viscous' in the revised manuscript. We ignore viscous stresses throughout the domain except through the wall model at the bottom wall. The subgrid (or sub-filter) scale stresses are active throughout the domain.

*- p. 6, line 131-33: Does equation 4 not give the deficit, rather than "mean velocity"? Please define $u_{tot}$*

**Response:** Eq. (4) gives the mean velocity, using the total deficit ($\Delta u_{tot}$) defined in eq. (3). There is no variable named $u_{tot}$.

*- p. 8, Fig. 3: Please be consistent in plotting. The linewidth in the symbols change from left to right, i.e. symbols are less clear.*

**Response:** The symbol linewidths are now uniform.

*- p. 11, line 195: "Grazing" appears a odd choice of word. Please rephrase.*

**Response:** This word has been replaced in the revised manuscript.

*- p. 15, line 158-261: Please rephrase these sentences. It does not appear as if turbine 3 in Fig. 11(f) produce "appreciable larger" power than for a single rotor.*

**Response:** The reviewer probably meant "Fig. 9(f)" and not "11(f)", and lines "258-261" and not "158-261" in his comment. We have rephrased these lines in the revised manuscript.

*- p. 15, line 268-270: The authors state "It is seen that P2-5 is larger for all 4-rotor wind farms...". This is not correct. If you look at Figure 10(c) there is actually a cross-over for the 3rd turbine, where the single rotor produces more. Be careful, when you do the aggregate statistics, because it gets lost. Please rephrase.*

**Response:** In Fig. 11 and the accompanying discussion, we talk about average of the relative powers of turbines 2 through 5 (aggregate statistics) and not individual turbine powers. The individual relative power values, and whether they are smaller or larger in the 4-rotor wind farms compared to the 1-rotor wind farms, have already been presented prior to this in Figs. 9 and 10 and associated discussion (lines 246-296), so we do not repeat the same observations here.

*- p. 16, Fig. 9: The axes on Fig. 9(a)+(d) seems wrong? If the velocity deficit is normalized, should the axes not be between 0 and 1?*

**Response:** The velocity deficits are normalized by the friction velocity of the precursor (ABL) simulation. So the values need not range from 0 to 1.

*- p. 17, Fig. 10: Please improve the figure. It is very difficult(impossible) to tell the lines apart as the symbols are so large that they cover the full vs broken lines. Comment on the cross-over at the 3rd turbine.*

**Response:** The figure has been modified. The differences between the lines are hopefully clear now. Comments on the crossover are added to Sec. 4.2 (lines 293-296).

*- p. 18, Fig. 11(b): Only three lines are visible. Please explain/comment in the text.*

**Response:** Three of the lines, corresponding to fixed $C'_T = 4/3$ and varying $S_X = 4D$, $5D$ and $6D$, lie on top of each other. This is included in the revised manuscript (lines 305-306).

*- p. 19, line 328: Typo. Correct to "reproduced".*

**Response:** The typo has been corrected.

*- p. 20, Fig. 13: The axes on Fig. 9(a)+(d) seems wrong? If the velocity deficit is normalized, should the axes not be between 0 and 1?*

**Response:** The reviewer probably meant "Fig. 13(a)-(d)" here. The velocity deficits are normalized by the friction velocity of the precursor (ABL) simulation. So the values need not range from 0 to 1.

*- p. 21-22, Fig. 14-15: Please include explanation of the red values and how they are computed. Figures should be self-contained.*

**Response:** The red values are the absolute error between the LES result and the model prediction of the relative power, computed at each of the four downstream turbines (turbines 2 through 5), and then averaged. Thus, the numbers in red in Figures 15 and 16 are $(1/4) \sum_{i=2}^{5} |(P_i/P_1)^{LES} - (P_i/P_1)^{model}|$. The formula for the error is included in the revised manuscript on line 375 and in the figure captions.

*Additional references:*

*1. van der Laan, M. P. and Abkar, M.: Improved energy production of multirotor wind farms, J. Phys.: Conf. Ser., 1256, 1–11, https://doi.org/10.1088/1742-6596/1256/1/012011, 2019.*

*2. Andersen and Ramos-Garcia: Dynamic Analysis of the Multi-rotor: Performance and Wake, WESC, 2019 https://doi.org/10.5281/zenodo.3357790*

**References**

N. S. Ghaisas, A. S. Ghate, and S. K. Lele. Large-eddy simulation study of multi-rotor wind turbines. *Journal of Physics: Conference Series*, 1037:072021, 2018.

A. Ghate and S. Lele. Subfilter-scale enrichment of planetary boundary layer large eddy simulation using discrete Fourier-Gabor modes. *Journal of Fluid Mechanics*, 819:494–539, 2017.

T. Nishino and R. H. J. Wilden. Effects of 3D channel blockage and turbulent wake mixing on the limit of power extraction by tidal turbines. *International Journal of Heat and Fluid Flow*, 37:123–135, 2012.

R. Stevens, D. Gayme, and C. Meneveau. Large eddy simulation studies of the effects of alignment and wind farm length. *Journal of Renewable and Sustainable Energy*, 6:023105, 2014.

M. P. van der Laan, S. J. Andersen, N. R. Garcia, N. Angelou, G. R. Pirrung, S. Ott, M. Sjoholm, K. H. Sorensen, J. X. Vianna Neto, M. Kelly, T. K. Mikkelsen, and G. C. Larsen. Power curve and wake analyses of the Vestas multi-rotor demonstrator. *Wind Energy Science*, 4:251–271, 2019.

Y.-T. Wu and F. Porté-Agel. Large-Eddy Simulation of Wind-Turbine Wakes: Evaluation of Turbine Parameterisations. *Boundary-Layer Meteorology*, 138: 345–366, 2011.

---

## Referee Report (RR1)

Second review comments for "Effect of tip spacing, thrust coefficient and turbine spacing in multi-rotor wind turbines and farms" by Ghaisas et al. in Wind Energy Science, 2019.

Thanks for your detailed response, and the added information, particular in the appendices. I only have a couple of follow-up comments.

1. Resolution and degree of detail.
- Thanks for the clarification in terms of grid points per rotor and the additional thoughts behind. However, I'm a little surprised that this has not been led to any changes in the article, but only explanations to me as a reviewer. I still believe this information should be stated explicitly in the article. Using established rule-of-thumbs are of course fine in general, but these rule-of-thumbs are for simulating single actuator discs. Actuator disc theory is based on 1D momentum theory, which comes with a number of assumption, e.g. the wake can expand freely afterwards. This is not the case in the multi-rotor. The combined induction effect of the multi-rotor is also different. My concern is essentially that some of the basic assumptions might be violated, hence general rule-of-thumbs are no longer valid. It is great that you have performed a grid convergence study. However, the difference in mean velocity(Figure 3) is not quantified, but it is discernible. If I zoom in and actually measure the difference for the red and blue grids in Fig. 3b) for x/D=6, I get an estimated difference of 5% in mean velocity, see attached sketch. Similar difference are seen in the other plots for the multi-rotor as well as the single rotor. If one simply assumes $P \sim U^3$, that would then correspond to a difference of $1.05^3 \sim 15\%$ in power. This appears to be comparable to the differences shown in Figure 9 and larger than your estimated errors between LES and model(Fig. 15-16), as well as the numbers reported in Appendix B. In general, your article would actually benefit from quantifying the results a bit more for better comparison. So to sum up. I understand that it is not necessarily feasible to perform the entire study on a fully converged grid and that a 5% difference in the mean velocity might be acceptable, if one is comparing results from the same numerical setup. However, the setup changes here and as previously mentioned the change in tip spacing are often less than the grid size. I believe it is good practice to discuss the limitations and possible violations of fundamental assumptions, so I encourage you to include these considerations in your article. It does not take anything away from your otherwise interesting results, on the contrary. It shows a cautious approach and critical sense of scientific results.

- My previous comment was: "The authors state "It is seen that P2-5 is larger for all 4-rotor wind farms...". This is not correct. If you look at Figure 10(c) there is actually a cross-over for the 3rd turbine, where the single rotor produces more. Be careful, when you do the aggregate statistics, because it gets lost. Please rephrase."
=> New comment: I understand that the data is aggregated, and the statement itself is not wrong. But it is a little "dangerous" to simply aggregate and conclude that the power production is larger for all multi-rotors wind farms(line 299-300) when the power difference is occassionally negative, i.e. sometimes it is better to have a single rotor. If you included more turbines the advantage migth disappear all together.

Finally, I also wish to point the authors attention to a newly published paper, which examines many of the same things and have comparable findings.

```
@article{bastankhah2019a,
    title = {Multirotor wind turbine wakes},
    language = {eng},
    publisher = {American Institute of Physics Inc.},
    journal = {Physics of Fluids},
    volume = {31},
    number = {8},
    pages = {085106},
    year = {2019},
    issn = {10897666, 10706631},
    doi = {10.1063/1.5097285},
    author = {Bastankhah, Majid and Abkar, Mahdi}
}
```

[Figure]

Fig. 3

(b) $\frac{x}{D} = 6$

$1 \text{m/s} \cdot \frac{5}{17} = 0.29 \text{ m/s}$

5 mm

$\sim 5.3 \text{m/s} \Rightarrow$

$\frac{0.29 \text{m/s}}{5.3 \text{m/s}} \approx 5\%$

17 mm

5.6 m/s

---

## Author Response (AR2)

**Responses to Reviewer Comments on "Effect of tip spacing, thrust coefficient and turbine spacing in multi-rotor wind turbines and farms"**

Niranjan S. Ghaisas, Aditya S. Ghate, Sanjiva K. Lele

**1  Response to Reviewer 2**

*1. Resolution and degree of detail.*

*- Thanks for the clarification in terms of grid points per rotor and the additional thoughts behind. However, I'm a little surprised that this has not been led to any changes in the article, but only explanations to me as a reviewer. I still believe this information should be stated explicitly in the article. Using established rule-of-thumbs are of course fine in general, but these rule-of-thumbs are for simulating single actuator discs. Actuator disc theory is based on 1D momentum theory, which comes with a number of assumption, e.g. the wake can expand freely afterwards. This is not the case in the multi-rotor. The combined induction effect of the multi-rotor is also different. My concern is essentially that some of the basic assumptions might be violated, hence general rule-of-thumbs are no longer valid. It is great that you have performed a grid convergence study. However, the difference in mean velocity(Figure 3) is not quantified, but it is discernible. If I zoom in and actually measure the difference for the red and blue grids in Fig. 3b) for x/D=6, I get an estimated difference of 5% in mean velocity, see attached sketch. Similar difference are seen in the other plots for the multi-rotor as well as the single rotor. If one simply assumes $P \sim U^3$, that would then correspond to a difference of $1.05^3 \sim 15\%$ in power. This appears to be comparable to the differences shown in Figure 9 and larger than your estimated errors between LES and model(Fig. 15-16), as well as the numbers reported in Appendix B. In general, your article would actually benefit from quantifying the results a bit more for better comparison. So to sum up. I understand that it is not necessarily feasible to perform the entire study on a fully converged grid and that a 5% difference in the mean velocity might be acceptable, if one is comparing results from the same numerical setup. However, the setup changes here and as previously mentioned the change in tip spacing are often less than the grid size. I believe it is good practice to discuss the limitations and possible violations of fundamental assumptions, so I encourage you to include these considerations in your article. It does not take anything away from your otherwise interesting results, on the contrary. It shows a cautious*

*approach and critical sense of scientific results.*

**Response:** We have performed simulations at two further levels of refinement, labeled G4 and G5 in the revised manuscript. These simulations have been performed for 1-rotor as well as 4-rotor turbine with $s/d = 0.05$. The results show that the velocity deficits and added turbulent kinetic energy (TKE) values are better converged at these finer resolutions. It is further observed that the convergence of velocity deficit profiles is non-monotonic, a common observation in channel flow simulations (Meyers and Sagaut, 2007). The convergence of the resolved TKE is monotonic, as expected. Comparing the results of our working grid G2 with grid G5, we obtain differences in velocity deficit of 3.2% and 1.9% at $x/D = 4$ and 6 respectively. In view of the marginal difference in the velocity deficits and the enormous added cost (almost 16 times), we feel that the use of grid G2 for evaluating the effect of all problem parameters is justified. We would like to reiterate that our grid size is in keeping with several previous studies as outlined in our previous response. Also, as mentioned by the reviewer himself, it is unreasonable to expect one to perform a parametric sweep on a fully converged grid. We also agree with the reviewer about the need to highlight possible limitations of a study. We have included all of the above information in the revised manuscript (lines 173 - 202). We hope this addresses the concerns of the reviewer and makes the article more useful to future readers.

*- My previous comment was: "The authors state "It is seen that P2-5 is larger for all 4-rotor wind farms...". This is not correct. If you look at Figure 10(c) there is actually a cross-over for the 3rd turbine, where the single rotor produces more. Be careful, when you do the aggregate statistics, because it gets lost. Please rephrase."*

*=¿ New comment: I understand that the data is aggregated, and the statement itself is not wrong. But it is a little "dangerous" to simply aggregate and conclude that the power production is larger for all multi-rotors wind farms(line 299-300) when the power difference is occassionally negative, i.e. sometimes it is better to have a single rotor. If you included more turbines the advantage migth disappear all together.*

**Response:** The text in lines 323 - 325 has been reworded to reflect the reviewer's concern.

*Finally, I also wish to point the authors attention to a newly published paper, which examines many of the same things and have comparable findings.*

**Response:** A reference to this paper is included in Section 1, lines 45 - 49, in the revised manuscript.

We thank Dr. S. J. Andersen once again for his careful assessment and hope to have addressed all concerns.

**References**

J. Meyers and P. Sagaut. Is plane-channel flow a friendly case for the testing of large-eddy simulation subgrid-scale models? *Physics of Fluids*, 19:048105, 2007.

[revised manuscript text omitted]